



# Uncertainty sources in simulated ecosystem indicators of the 21st century climate change

Jarmo Mäkelä[1], Francesco Minunno[2], Tuula Aalto[1], Annikki Mäkelä[2], Tiina Markkanen[1], and Mikko Peltoniemi[3]

[1]Finnish Meteorological Institute, P.O. Box 503, FI-00101 Helsinki, Finland
[2]Department of Forest Sciences, P.O. Box 27, FI-00014 University of Helsinki, Finland
[3]Natural Resources Institute Finland, P.O. Box 2, FI-00791 Helsinki, Finland

**Correspondence:** Jarmo Mäkelä (jarmo.makela@fmi.fi), Francesco Minunno (francesco.minunno@helsinki.fi)

**Abstract.** The forest ecosystems are already responding to increased $CO_2$ concentrations and changing environmental conditions. These ongoing developments affect how societies can utilise and benefit from the woodland areas in the future, be it e.g. climate change mitigation as carbon sinks, lumber for wood industry or preserved for nature tourism and recreational activities. We assess the effect and the relative magnitude of different uncertainty sources in ecosystem model simulations from the year

1980 to 2100 for two Finnish boreal forest sites. The models used in this study are the land ecosystem model JSBACH and the forest growth model PREBAS. The considered uncertainty sources for both models are model parameters, four prescribed climates and two RCP (Representative Concentration Pathway) scenarios. PREBAS simulations also include an additional RCP scenario and two forest management actions. We assess the effect of these sources at four different stages of the simulations on several ecosystem indicators of climate change, e.g. gross primary production (GPP), ecosystem respiration, soil moisture,

recurrence of drought, length of the vegetation active period (VAP), length of the snow melting period and the stand volume. The climate model uncertainty remains roughly the same throughout the simulations and is overtaken by the RCP scenario impact halfway through the experiment. The management actions are the most dominant uncertainty factors for Hyytiälä and as important as RCP scenarios at the end of the simulations, but contribute only half as much for Sodankylä. The parameter uncertainty is the most elusive to estimate due to non-linear and adverse effects on the simulated ecosystem indicators.

## 1  Introduction

The global atmospheric greenhouse gas concentrations are rising and inducing changes in land ecosystem carbon balances, water cycles and their seasonality. The rate of the expected concentration rise depends on human actions and the corresponding emission pathways chosen. The pathways presented in IPCC AR5 report (IPCC, 2014) lead to a radiative forcing of 2.6 W/m$^2$ to 8.5 W/m$^2$ in the year 2100. In addition to climate pathways connected to human actions, the variability in the IPCC climate

projections is due to model differences and to internal variability in the climate system. Climate sensitivity has proven to be extremely difficult to constrain (Knutti and Sedláček, 2012). The multi-model spread in e.g. temperature and precipitation has not been narrowing during the last few years despite substantial model development (Eyring et al., 2019). However, narrowing the uncertainties should not be the only aim and sign of progress in climate modelling. Models improve as more processes





are described in detail, which may also introduce new unknown uncertainties. Thus it is important to study what are the

contributions of different factors to the total uncertainty of examined variables, and how does the uncertainty evolve in the future.

The climate models provide drivers for the land ecosystem models. The predictions by land ecosystem models are affected by the driver uncertainties and by uncertainties related to the land surface model itself. Usually, only variability between different models is examined (see e.g. Friend et al., 2014; Nishina et al., 2015), and the uncertainty related to model parameters

is not taken into account (Reyer et al., 2016). The unaccounted model processes can lead to significant underestimation of the overall uncertainty (Trugman et al., 2018). Furthermore, the spread in the uncertainty of the model outcome depends on the variable and region investigated. High latitude ecosystems are predicted to experience significant changes due to climate warming (Schaphoff et al., 2015). The change in seasonality of the ecosystems is predicted to manifest itself via decrease in snow cover duration, earlier soil thaw and later soil freeze and longer growing season (Dye and Tucker, 2003; McDonald

et al., 2004; Barichivich and Caesar, 2012). The longer growing season and warmer temperatures are predicted to increase both ecosystem carbon uptake and respiration (Piao et al., 2008), while harmful extremes connected to heat, soil drought and soil excess water are also predicted to become more severe (Ruosteenoja et al., 2017). The evolution of net ecosystem exchange (NEE), defined as the difference between net ecosystem primary production (NPP) and heterotrophic respiration ($R_h$), is rather uncertain in future due to opposing drivers and may follow a trend towards net emissions or net uptake.

Forest management in Finland is a strong modifier of ecosystem carbon budgets and usually an unaccounted source of uncertainty in future predictions. The harvesting intensity defines the impact to the ecosystem carbon exchange (Korkiakoski et al., 2018). According to Kalliokoski et al. (2018), the future forest productivity was predicted to increase towards the end of the century. The climate model ensemble predictions were the dominant source of uncertainty for forest productivity, but closer to the end of century the role of emission pathways became more important. Estimation of future development of ecosystem

carbon budgets together with impact factors such as management, seasonality and water conditions adds information to the whole ecosystem functioning. Assessment of uncertainties related to carbon budgets and growing season length together with water and snow conditions is important in estimating the forests ability to provide ecosystem services related to e.g. carbon sequestration, wood harvesting, maintaining habitats and promoting nature tourism (Snell et al., 2018; Holmberg et al., 2019).

Here we estimate how biomass, carbon, growing season, water and snow -related ecosystem indicators of climate change

and their uncertainties progress in the future. We engage two ecosystem models at southern and northern boreal forest sites – JSBACH is developed to study land surface processes with closely coupled carbon balances and hydrology, while PREBAS is aimed to study carbon budgets with implementation of forest management. Both models have been previously calibrated for boreal ecosystems (Mäkelä et al., 2019; Minunno et al., 2019). We estimate the contribution of model parameter uncertainty, climate model variability, RCP pathway and management actions to the total uncertainty of these indicators. We apply canonical

correlation analysis (CCA) to cross-correlate the uncertainty sources with the chosen ecosystem indicators. Finally, we aim to combine the model estimates to determine which are the dominant sources of uncertainty in future ecosystem projections.





## 2  Materials and methods

We will first briefly introduce the sites and their characteristics, followed by the RCP scenarios and climate models used in this study as well as the models used to run the simulations. Next we describe our ecosystem indicators of climate change and
define the methods used to analyse the simulations.

### 2.1  Sites

The sites used in this study are called Hyytiälä (FI-Hyy; 61°51′N, 24°17′E, 180 m a.s.l.) and Sodankylä (FI-Sod; 67°22′N, 26°38′E, 179 m a.s.l.); they are respectively located in southern and northern Finland and represent the southern and northern boreal pine forests. These sites can be characterised as Boreal evergreen needleleaf forests, where the dominant species is the
Scots pine (*Pinus sylvestris*).

The Hyytiälä site (Kolari et al., 2009) was planted in 1962, after burning and mechanical soil preparation. The soil type is Haplic Podzol on glacial till. The site has an understory of Norway spruce (*Picea abies*) and few deciduous trees. The maximum measured all-sided leaf area index (LAI) for the Scots pine is 6.5 m$^2$/m$^2$, the average measured annual precipitation is 709 mm and temperature 2.9 °C.

The Sodankylä site (Thum et al., 2007) has been naturally regenerated after forest fires and hosts trees ranging from approximately 50 to 100 years of age. The soil type is fluvial sandy Podzol. The ground vegetation consists of lichens, mosses and ericaceous shrubs. The maximum measured LAI for the Scots pine is 3.6 m$^2$/m$^2$, as determined from forest inventories, the annual precipitation is 527 mm and temperature -0.4 °C.

### 2.2  RCP scenarios and climate models

We selected model runs of the fifth phase of the Coupled Model Intercomparison Project (CMIP5; Meehl et al., 2009; Taylor et al., 2012) following three representative concentration pathway (RCPs), that reach radiative forcing levels of 2.6, 4.5 and 8.5 W/m$^2$ by the end of the century (Moss et al., 2010; van Vuuren et al., 2011). Throughout the historical period that ends in 2005 the land cover data and the greenhouse gas concentrations corresponding different RCPs follow common trajectories (Meinshausen et al., 2011).

Climate data for years 1980-2100 was obtained from five global climate models (GCMs; CanESM2, CNRM-CM5, GFDL-CM3, HadGEM2-ES and MIROC5). The climate variables were bias corrected and further down-scaled to a 0.2°×0.1° longitude-latitude grid, similarly to Lehtonen et al. (2016); Holmberg et al. (2019). The bias correction methods are described in Räisänen and Räty (2013); Räty et al. (2014). The harmonised FMI meteorological data by Aalto et al. (2013) was used as reference.

The sub-set of five climate models was selected because of their good performance in reproducing current climate in Northern Europe and because they provided complete data sets for running impact models (Lehtonen et al., 2016). The five chosen models represent well the variation from current climate conditions (1981-2010) to the end of the ongoing century (2070-2099). The winter-time (i.e. December, January and February) precipitation in Finland for the five models in RCP4.5, covers





the range of variability depicted by 24 out of 28 CMIP5 models investigated by Ruosteenoja et al. (2016). In summer the

precipitation change range is generally narrower than in winter and the selected models cover the range of roughly half of the 28 CMIP5 models. Winter temperature change shows intermediate values among the 28 models and the range captures the ranges of change shown by 11 models. In summer the five model selection represents the range of change depicted by the upper half of the 28 models analysed by Ruosteenoja et al. (2016). Furthermore, the five climate models represent host institutes from different countries and from three continents: Asia, Europe and North-America. $CO_2$ concentrations from the RCPs 2.6, 4.5

and 8.5 increased monotonously through the calendar years reaching respective global means of 421, 538 and 936 ppm by the end of the century. PREBAS was run with results from all five climate models and three RCP scenarios, whereas JSBACH simulations included only RCP4.5 and RCP8.5 due to missing bias corrected climate variables. Moreover and for the same reason, JSBACH was not run with the HadGEM2-ES climate model for RCP8.5.

## 2.3 The JSBACH model

The JSBACH ecosystem model (Kaminski et al., 2013) is the land-surface component of the Earth system model of the Max Planck Institute for Meteorology (MPI-ESM). In these simulations, the model setup and parameter distributions are derived from Mäkelä et al. (2019). JSBACH is used uncoupled from the atmosphere, applying five layers within a multilayer soil hydrological scheme (Hagemann and Stacke, 2015) and utilising the BETHY model for canopy/stomatal conductance control (Knorr, 2000). Additionally, the model effectively uses only one plant functional type (PFT), coniferous evergreen trees.

The JSBACH model uncertainty is represented by a set of 100 parameter vectors, defined and described in more detail in Appendix A. The parameter distributions were derived from the simulations described in Mäkelä et al. (2019), where the model is calibrated and validated with site level measurements from 10 different evergreen needleleaf forests throughout the boreal zone (including Hyytiälä and Sodankylä). In order to avoid confusion with the climate models, the model uncertainty will be henceforth referred to as parameter uncertainty.

The JSBACH model initial state was derived from the end state of several thousand year long regional simulations that equilibrate the soil carbon storages. In addition, the simulations included a simulation specific spin-up period of 20 years to ensure adequate site level LAI and soil water storages. The spin-up was achieved by running the model through the first 20 years of simulation data, saving the state of the model variables and using them as the initial state for the 120-year long simulations. This type of spin-up introduces a discontinuity between the initial state and the driving climate but differences in

the examined climate indicators should be negligible.

## 2.4 The PREBAS model

PREBAS (Valentine and Mäkelä, 2005; Peltoniemi et al., 2015; Minunno et al., 2019) is a simplified forest carbon and water balance model, which also considers forest growth and management. It calculates photosynthesis (GPP) using a light-use-efficiency (LUE) approach and ambient $CO_2$ concentration (Peltoniemi et al., 2015; Minunno et al., 2016). Daily GPP is

influenced by soil moisture, radiation, temperature, vapour pressure deficit and precipitation. The model also calculates evapotranspiration (ET) and updates the water balance daily. Mean tree growth is calculated from GPP and respiration at an annual





time step, and growth is allocated to different tree organs under assumptions on tree structure (Valentine and Mäkelä, 2005). The model includes tree mortality due to crowding. The growth module annually updates the canopy leaf area index (LAI) for the GPP and ET estimation. In order to estimate soil carbon, the annual litter fall is calculated by the growth allocation module, and fed to Yasso07 soil carbon model (Liski et al., 2005; Tuomi et al., 2009). NEE is calculated annually.

In addition to weather data, PREBAS requires information about the initial state of the simulated forest, defined as soil fertility class, stand basal area, mean height and mean diameter, at an appropriate spatial resolution. This information was extracted from the multisource forest inventory data maps (Tomppo et al., 2014; Mäkisara et al., 2016). The forest resource maps have a 16 m resolution and report the forest data for the year 2015. The model was initialised with forest data extracted for an area of $8 \times 8$ km square centered at the eddy covariance towers of Hyytiälä and Sodankylä.

In this study, two management scenarios were used in PREBAS simulations. The business as usual (BAU) scenario follows present forest management recommendations in Finland (Rantala et al., 2011), where trees have to be at least 24–30 cm diameter at breast height (dbh; 130 cm) and of age from 60–100 years before harvesting. The delayed ecosystem logging (DEL) scenario aims for the near term carbon sink increase by increasing the minimum harvesting diameter to 36 cm dbh.

## 2.5 Ecosystem indicators of climate changes

We study the uncertainty sources related to key biophysical indicators and their future development. Thus we ran the JSBACH and PREBAS models with different combinations of climate, RCP and management (only for PREBAS) scenarios with each realisation of the model parameterisations, resulting in approximately 2 000 site specific simulations for JSBACH and 6 000 for PREBAS. These simulations produced daily variables that were used to calculate the ecosystem indicators of climate change, presented in Table 1. We have included details on how we calculated the derived variables (number of dry days, start and end days of growing season and snow melting period) in Appendix B.

## 2.6 Analysis of results

We analyse the results by producing means, standard deviations and correlations of the model variables. This analysis is based on the annual values or averages over certain months (e.g. summer soil water) – one value per year. We utilise the Mann-kendall test (Mann, 1945; Kendall, 1975) to verify the existence of trend lines and kernel density estimation (KDE) to visualise the distribution of values (this approach can be viewed as a smoothed histogram).

We also carried out canonical correlation analysis (CCA) to quantify the impact of the different factors on the ecosystem indicators. The factors in this analysis are parametric uncertainty (par), climate models (clim) and RCP scenarios (rcp) for JSBACH and additionally management scenarios (man) for PREBAS. The indicators were averaged and divided into four consecutive 30-year long periods for both models: 1980-2009 (reference), 2010-2039 (interim), 2040-2069 (mid-century) and 2070-2099 (future). This produced single indicator values for each period and simulation (single instance of each factor) that were calculated for both sites separately.

CCA is a multivariate extension of correlation analysis that allows identifying linear relationships between two sets of variables (Hotelling and Pabst, 1936). We summarise the CCA results with the use of the redundancy index ($Rd$) that expresses





**Table 1.** Ecosystem indicators derived from the recorded values of the JSBACH and PREBAS simulations, separated into groups for the canonical correlation analysis. The group names relate to biomass distribution, ecosystem carbon exchange, length of the growing season, water cycle and snow melting period.

| Indicator | Abb. | Units | JSB | PRE | Group |
|---|---|---|---|---|---|
| basal area | BA | $m^2$ / ha | | x | Biomass |
| stand volume | V | $m^3$ / ha | | x | Biomass |
| harvested volume | Vharv | $m^3$ / ha | | x | Biomass |
| volume of dead trees | Vmort | $m^3$ / ha | | x | Biomass |
| tree biomass | Biom | kg(C) | | x | Biomass |
| tree litterfall | Lit | kg(C) | | x | Biomass |
| leaf area index | LAI | $m^2$ / $m^2$ | | x | Biomass |
| gross growth | Growth | kg(C) / year | | x | Biomass |
| gross primary production | GPP | g(C) / $m^2$ day | x | x | Carbon |
| net primary production | NPP | g(C) / $m^2$ day | x | x | Carbon |
| net ecosystem exchange | NEE | g(C) / $m^2$ day | x | x | Carbon |
| respiration (at) | Raut | g(C) / $m^2$ day | x | x | Carbon |
| soil carbon | Csoil | kg(C) | | x | Carbon |
| start of growing season | SOS | DOY | x | x | Growth |
| end of growing season | EOS | DOY | x | x | Growth |
| length of growing season | VAP | days | x | x | Growth |
| evapotranspiration | ET | mm / day | x | x | Water |
| annual soil water | aSW | mm | | x | Water |
| summer soil water | sSW | mm | x | x | Water |
| number of dry days | Ddry | days | x | | Water |
| albedo | alb | | x | | Snow |
| snow amount | snow | m | x | | Snow |
| start of snow melt | melt | DOY | x | | Snow |
| snow clear date | clear | DOY | x | | Snow |
| length of snow melt | SM | days | x | | Snow |

the amount of variance of a set of variables explained by another set of variables (Stewart and Love, 1968; Weiss, 1972; van den Wollenberg, 1977). The details of the CCA and the redundancy index are given in appendix C.



## 3  Results

Forest management was the most dominant factor of uncertainty for Hyytiälä (Fig. 1) throughout the simulation. There was a clear difference for Sodankylä, where management gains only half as much influence. Disregarding management, the climate

models and RCP scenarios represent major sources of both JSBACH and PREBAS predictive uncertainty. The impact of climate models was dominant during the reference and interim periods and remained roughly constant over time. The importance of RCP scenarios increased towards the end of the simulations, catching up to management impact at Hyytiälä in mid-century and representing the most important factor during the last period. The parametric uncertainty was the least influential factor for both JSBACH and PREBAS, at both sites.

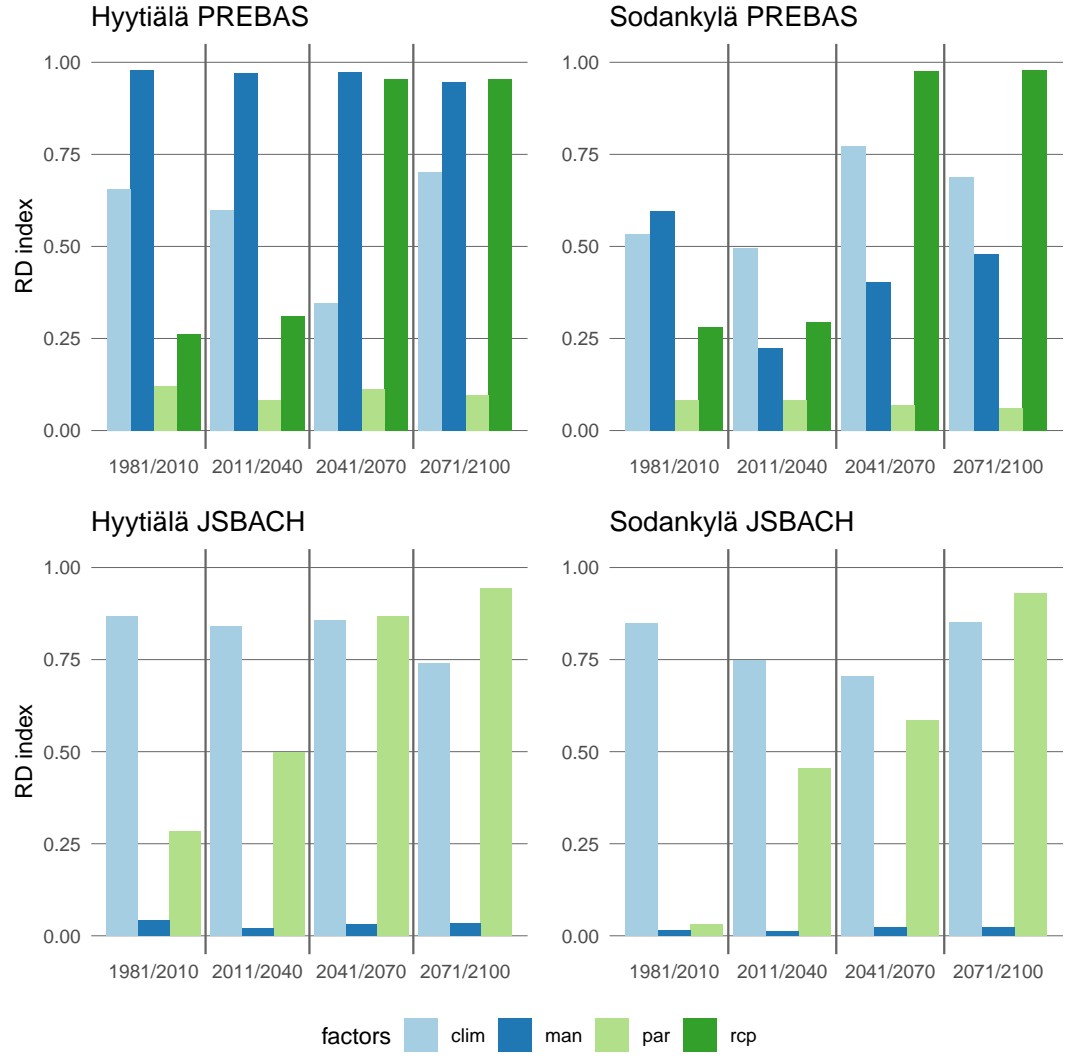

**Figure 1.** Redundancy indices calculated using all ecosystem indicators.



**Figure 2.** Redundancy indices calculated separately for the different indicator groups.





## 3.1 Biomass distribution

The site-level differences in biomass stock uncertainties largely arise from the management actions (Fig. 2) and the management and RCP scenario impacts reflect the redundancy indices calculated with all ecosystem indicators (Fig. 1) for PREBAS. The RCP scenario influence increases for both sites towards the end of the simulations and the climate model and parameter uncertainty is negligible for both sites and all periods. There is an anomaly for Sodankylä reference period, where management has a very large impact. This situation arises due to minimal (0.1 m$^3$/ha), but systematic difference in harvested volume – the difference is so small it is not visually evident (Fig. 3). The rest of the Sodankylä reference period variables are nearly identical, so the small change in harvesting results in high correlation, which is captured by the CCA.

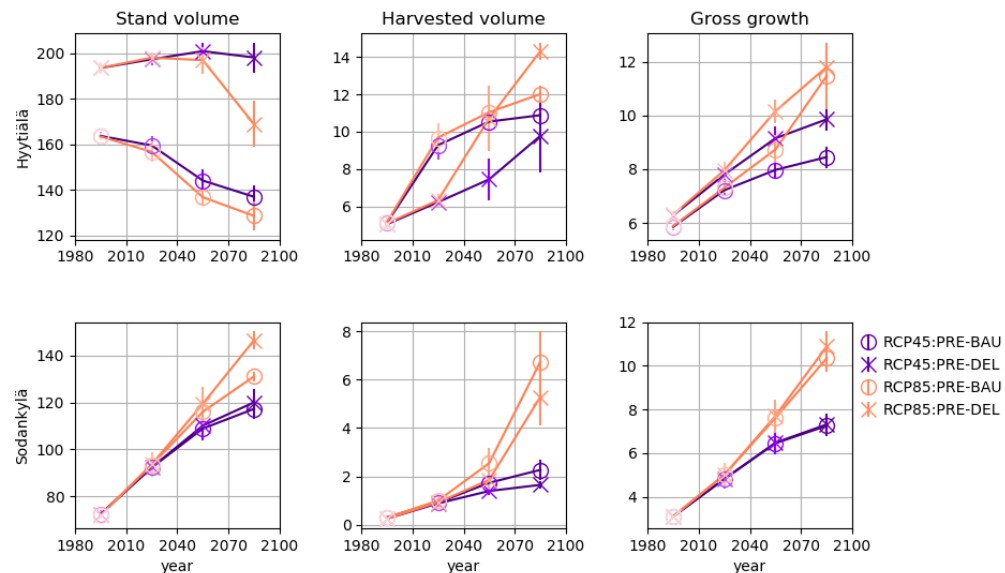

**Figure 3.** Selected ecosystem indicators from the PREBAS biomass factors, averaged for the 30-year long periods. The y-axis "whiskers" at each point represent the point specific uncertainty: one standard deviation amongst the corresponding simulations. We use lighter shading for the earlier periods, a different colour for the RCP scenarios and a different marker to separate the management actions.

The differences in site-specific variables due to the management actions, can already be seen from the reference period indicators (Fig. 3). The DEL scenario has approximately 10 % larger stand volume than BAU for Hyytiälä, but there is practically no difference for Sodankylä. The management actions start to have a noticeable impact for Sodankylä simulated variables at mid-century, but this impact is much smaller than that of the RCP scenarios. The management effect is much more pronounced at Hyytiälä, where both actions follow separate pathways.





## 3.2 Ecosystem carbon exchange

The bifurcation of the annual GPP and respiration in JSBACH illustrates the separation of the RCP scenarios at about the
midpoint (2040) in the simulations (Fig. 4). These two variables that comprise the net ecosystem exchange (NEE), have strong
temporal linear correlations for both RCP scenarios ($r2 \approx 0.95$). The respective linear regression lines for GPP [g(C)/m$^2$d]
yield an increase of 1.3 and 2.4 (RCP4.5 and 8.5) in 100 years for Hyytiälä and similarly 0.6 and 0.8 for Sodankylä. Likewise,
the increases in respiration are 1.6 and 2.6 for Hyytiälä in 100 years and 0.8 and 1.2 for Sodankylä. GPP uncertainty was larger
at the beginning of the simulations, but levelled with respiration at the end of the period. Relatively, the increased radiative
forcing yields a stronger increase in GPP for Hyytiälä and respiration for Sodankylä. Some of the flux variables, such as
Sodankylä GPP (Fig. 4), suggest a bi-modal value distribution in the the last 30 years of the simulations. This is caused by the
different climate models yielding separate modes to the otherwise nearly identical value distributions. Most of the GPP and
respiration value distribution (Fig. 4) reflect the variation in model parameterisations.

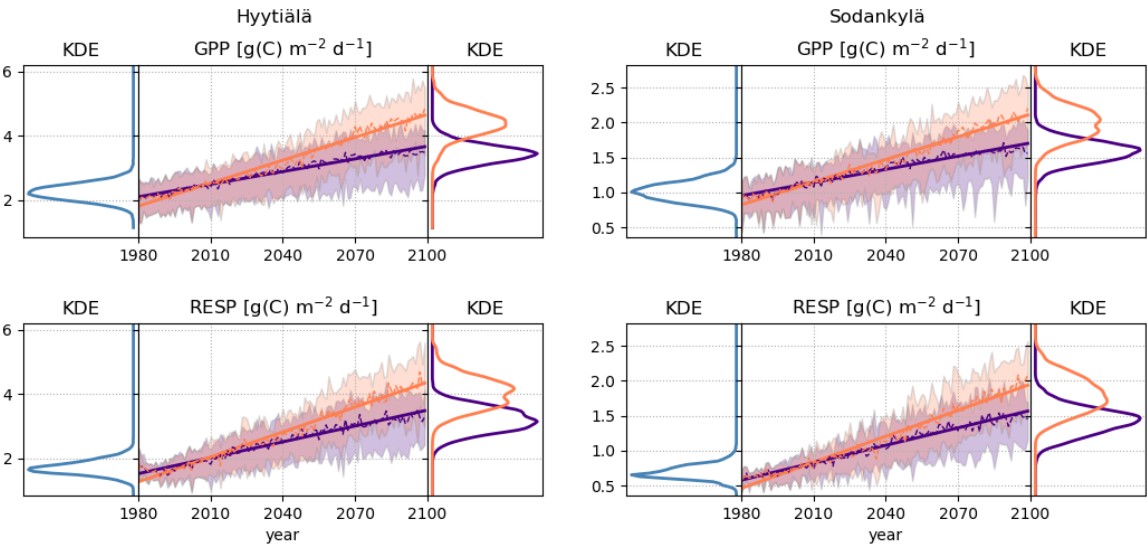

**Figure 4.** JSBACH predicted annual values of GPP and respiration for RCP4.5 (purple) and RCP8.5 (orange) scenarios. The shaded area
represents all RCP-specific simulations, the dashed line is the annual mean and the solid line is the trend line. The KDE estimates on the left
side of each image represents the distribution of the reference päeriod values of both RCP scenarios (blue), whereas the KDE on the right
side consists of RCP specific values from the last 30 years of simulations.

As the bifurcating GPP and respiration fluxes signal, the RCP scenarios were important sources of uncertainty for the
ecosystem carbon exchange variables at both sites, with importance growing over time (Fig. 2). However, it is noteworthy
that management induced uncertainty for ecosystem carbon exchange was the most influential factor for Hyytiälä when it is
accounted for in the model. The Sodankylä flux variation seems to be only dependent on the RCP scenario for both models,
while the climate models were the most important factors at Hyytiälä during the first two periods for JSBACH.





### 3.3 Ecosystem seasonality

The seasonal indicators depict the length of the vegetation active period and the snow melting period as well as the amount of soil water (and the recurrence of summer drought). The CCA analysis (Fig. 2) indicates that growing season indicators respond to changes in both climate models and RCP scenarios for both models, but the indicators are not sensitive to management actions. The snow melting period uncertainty for JSBACH is dominated by the climate models for the first half of the simulations for Hyytiälä, after which the RCP scenario is more influential. The situation is a bit different for Sodankylä snow melt, where

the climate model uncertainty reduces radically after the reference period and then remains the same – the RCP scenarios gain effectiveness as simulations progress and reach the climate model influence at mid-century. The uncertainty related to the water balance for JSBACH is not explained by any of the examined factors and the uncertainties for PREBAS are also low.

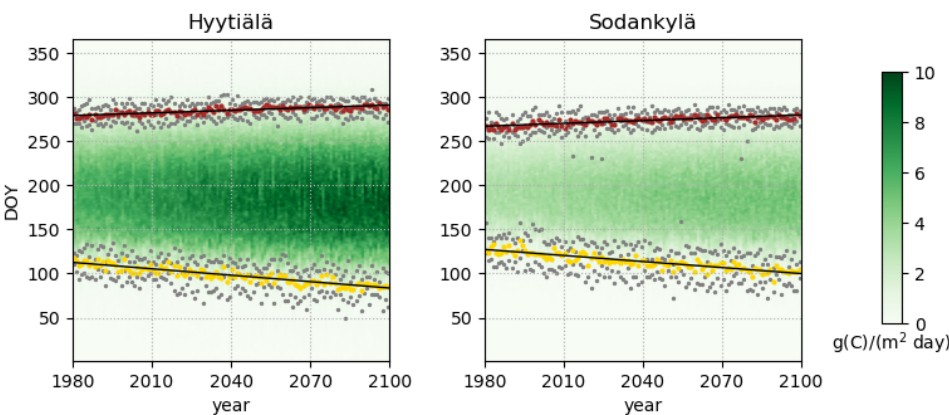

**Figure 5.** Average vegetation active period for JSBACH RCP4.5; yellow dots are the SOS values, red dots are the EOS values and the grey dots are the minimum and maximum SOS/EOS from all simulations. Also presented are the trend lines and the daily GPP as the green amplitude.

The vegetation active period is lengthening at both sites (Fig. 5). The displacement of the trendline start of (vegetation active) season (SOS) for JSBACH is approximately -8.1 days in 100 years for Hyytiälä (-11.3 for RCP8.5) and -7.6 days for

Sodankylä (-10.9). Likewise, the end of season (EOS) displacement is 3.3 days for Hyytiälä (5.1 for RCP8.5) and 3.5 days for Sodankylä (5.2). The SOS and EOS temporal correlations are typically strong ($r^2 \approx 0.8$). The increase to the length of VAP is very similar for both sites, regardless of the different annual GPP.

The Mann-Kendall tests report a decreasing trend (earlier occurrence) for start of the snow melting period, first snow-free date and the length of the snow melting period (Fig. 6) in all simulations, except for Sodankylä RCP8.5 where the Mann-

Kendall signifies the absence of trend for the melting period length. The simulations indicate that at the end of the century, the annual amount of snow in Hyytiälä will be radically diminished, and that Sodankylä winters will be similar to present



day Hyytiälä winters (especially in the RCP8.5 scenario). Relatively, the first snow free date is catching up to the start of the snow melting period (Fig. 6). The snow starts to melt approximately 20.7 days earlier in 100 years time for Hyytiälä RCP4.5 and 24.9 days earlier in RCP8.5, whereas the snow free dates appear 29.8 days (RCP4.5) and 41.7 days (RCP8.5) earlier.

The corresponding values for Sodankylä are 12.2 (RCP4.5) and 25.1 (RCP8.5) for the start of snow melting period and 20.0 (RCP4.5) and 28.2 (RCP8.5) for the snow free dates. The correlations vary widely: $r^2 \approx 0.7$ for snow free dates, $r^2 \approx 0.5$ for the start of the melting period and $r^2 \approx 0.2$ for their difference.

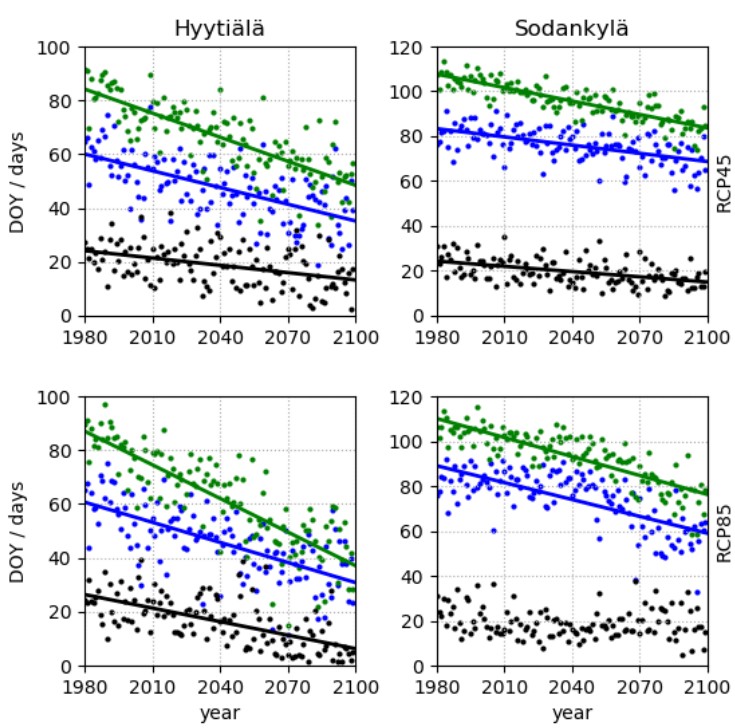

**Figure 6.** The average snow melting period for the JSBACH model; presented are the average annual values for the start of the snow melting period (blue), the first snow free day of the year (green) and their difference (black) as well as trend lines (calculated from the shown values) for these variables (when applicable).

The initial distributions of the summertime soil moisture values (Fig. 7) are unimodal for Hyytiälä and bimodal for Sodankylä for all climate models. This structure is still evident for the RCP4.5 scenario (of the last 30 years) but breaks down for the

RCP8.5. Moreover, Hyytiälä RCP8.5 demonstrates some bimodality for two of the climate models whereas the RCP8.5 for Sodankylä seems to be losing the bimodality and is becoming (in appearance) more similar to the Hyytiälä reference period. The model parameterisations result in highly similar soil moisture distributions for the reference period, but there are clear differences (distribution modes and shapes) for the last 30 years.



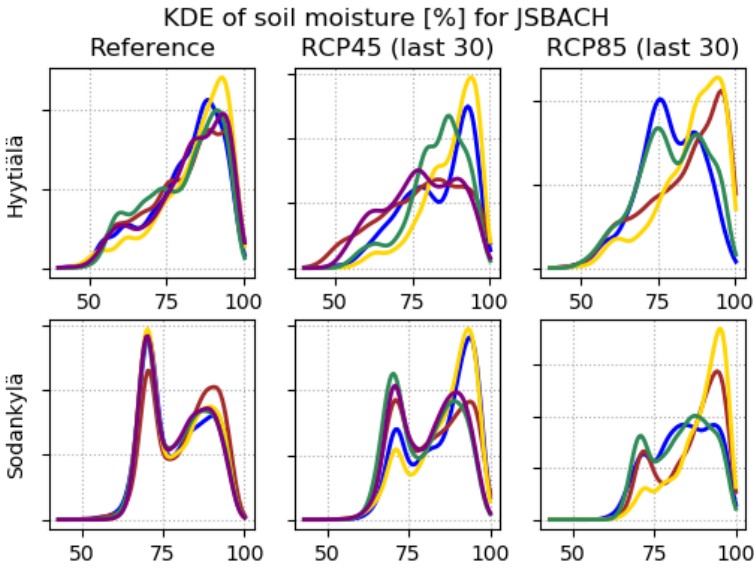

**Figure 7.** KDE estimates of the JSBACH soil moisture values (relative to soil field capacity) for the reference period and the last 30 years of simulations. Each colour represents the average summertime (June-August) soil moisture, produced with one of the climate models using all parameterisations.

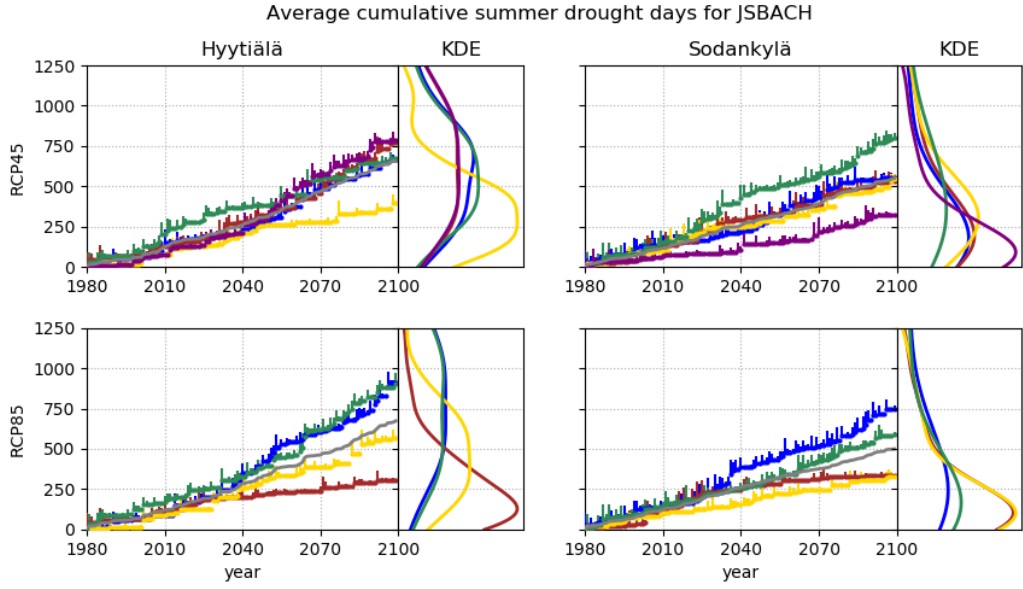

**Figure 8.** Accumulated summer drought days scatter plotted for each climate model, averaged over model parameterisations with minimum and maximum increment visualised as y-axis whiskers. The gray line is the average of the simulations. The KDE estimates on the right side depict the distribution of the accumulated drought days with the different parameterisations at the end of the simulation. The KDE figures have been cut at 1250 accumulated days.





The averaged drought events (Fig. 8) seem to be repeating at a roughly constant rate although the different model param-
eterisations result in wide soil moisture distributions (Fig. 7) at the end of the simulations. The average cumulative values
correspond reasonably well with the drought indicator threshold in B1 (5 % of 92 summertime days, accumulated for 120
years would result in 552 days). The temporal correlations for the individual climate model and RCP specific simulations is
poor ($r^2$ ranging from 0.12 to 0.5). The cumulative drought day distributions at the end of the simulations are strongly skewed
with wide "tails" and high-value outliers (outside the figures) of approximately 2600 drought days for Hyytiälä and 3700 for
Sodankylä. Interestingly, one of the climate models (brown) markedly reduces the amount drought days for the RCP8.5 at both
sites when compared to RCP4.5. Neither the accumulated drought day variations or those of the soil moisture values (Fig. 7)
are reflected in the CCA analysis of the Water group (Fig. 2)

### 3.4  Ecosystem indicator value comparison

The comparison results (Fig. 9) for soil moisture and ET indicate very small changes in the average values for both mod-
els but the JSBACH simulations manifest substantially larger variation. The JSBACH model yields more elevated levels of
relative GPP, NPP, NEE and respiration for Hyytiälä, but the situation is (mostly) reversed for Sodankylä. These differences
likely reflect the effect of the management actions and distinct site characteristics. The managements result in clearly different
pathways for these variables at Hyytiälä, but only yield small differences at the end of the simulation for Sodankylä.

The SOS is roughly identical for both models, whereas both PREBAS versions have a larger effect on the EOS – initially the
EOS for PREBAS occurs much earlier (roughly 15 days) than for JSBACH, which is diminished to a few days at the end of the
simulations. The PREBAS extends the VAP more evenly from both "ends", whereas JSBACH focuses more on the SOS. These
differences are reflected in the length of the VAP, which is merely the difference between EOS and SOS. Additionally, we note
that the largest value spreads (deviations as represented by the length of the "whiskers") appear during the values representing
the last 30 years of the RCP8.5 simulations – this merely reflects that the simulation uncertainties are increasing towards the
end of the simulation (as expected). Overall, the model responses to the different inputs is very alike, which results in linear
dependencies between the variables (Fig. 9).

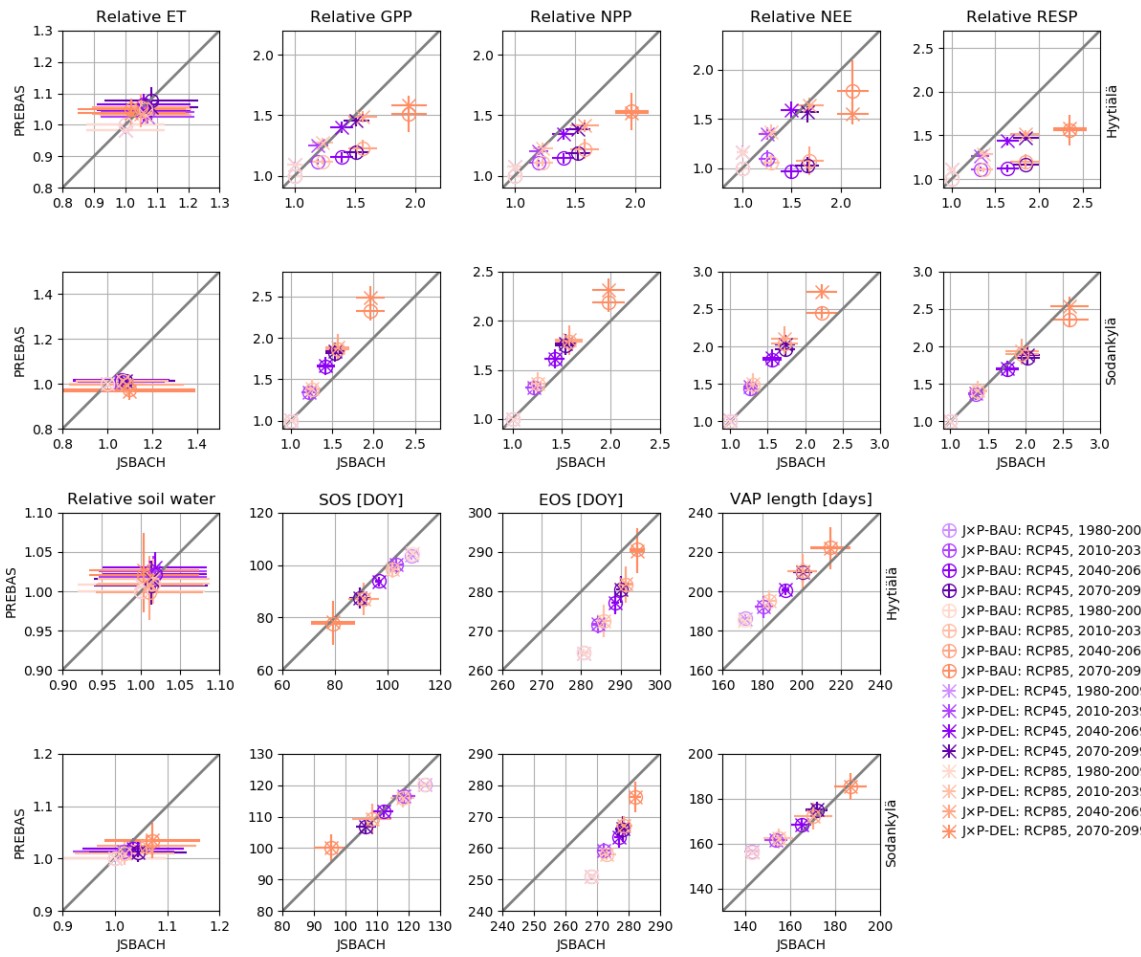

**Figure 9.** Average simulated values for shared ecosystem indicators between JSBACH and PREBAS, plotted for each 30-year period and both RCP4.5 and RCP8.5 scenarios. The values for JSBACH are divided by the average of the reference period values, and the values for PREBAS by the average of the BAU scenario reference period values. The "whiskers" at each point represent the point specific uncertainty: one standard deviation amongst the corresponding simulations. We use lighter shading for the earlier periods.





## 4 Discussion

In this paper we present an assessment on the importance of the different uncertainty sources, simulated on boreal forests for the 21st century. The JSBACH and PREBAS models yield similar uncertainty estimates (Fig. 1) and have a similar response to many of the examined ecosystem indicators of climate change (Fig. 9). The differences in modelled variables can be explained by the different model structures (e.g. soil moisture and evapotranspiration) or the inclusion of PREBAS management actions (ecosystem carbon fluxes). Forest management plays an important role in the estimates of ecosystem variables and their uncertainties. This importance is underscored by the lack of management in many land-surface components of climate models.

### 4.1 Impact to ecosystems

According to Grönholm et al. (2018), the long-term eddy-covariance measurements (1997–2017) at a boreal coniferous forest in Hyytiälä indicate a significant increase of gross-primary productivity (+10.5 [g(C)/m$^2$ year]), which is only partly compensated by an increased ecosystem respiration (+4.3g [g(C)/m$^2$ year]). As a result, the annual $CO_2$ sink has increased by about 6.2 [g(C)/m$^2$ year]. The GPP increase is dominated by an increase in LAI (from 4.1 to 4.6), while rise in the atmospheric CO2 concentration (from 360 ppm to 410 ppm) contributes only about 10 % to the rising GPP trend. It has to be noted that Hyytiälä forest was thinned in 2002, temporarily reducing LAI to 3.4. However, in few years the forest recovered to similar steadily increasing LAI trend than before thinning. The observed rise in the GPP is better replicated by the RCP8.5 scenario (Fig. 4) that yields an increase of +8.8 [g(C)/m$^2$ year] for Hyytiälä; whereas the increase in respiration is more closely reproduced by the RCP4.5 scenario (+5.8).

The RCP scenarios have a strong impact for growing stock and wood harvesting (Fig. 3), but the effect pales in comparison to the examined management actions. This underlines the importance of proper forest management for provisioning services (Snell et al., 2018; Holmberg et al., 2019). This is illustrated by the relative NEE pathways (Fig. 9) that are roughly convex for BAU and concave for DEL management actions. The simulations also indicate linearly lengthening VAP (Fig. 5), with high variation towards the end of the simulations (Fig. 9). This can be interpreted as beneficial for nature tourism and recreational activities, but on the other hand are the adverse effects of shortened snow melting period (Fig. 6) and potentially increased droughts (Fig. 8), also investigated by Ruosteenoja et al. (2017). These effects are also detrimental for winter harvesting and wood quality, as suggested by Holmberg et al. (2019).

Manninen et al. (2019) reported lengthened snow melting periods for some regions in Finland for 1982–2016. We analysed the reference period (1980–2009) snow melt in more detail and found that roughly 30 % of parameter specific simulations for Hyytiälä, and 20 % for Sodankylä, resulted in increased length for the snow melting period. We note that our simulations are restricted to site level, whereas regional experiments include lakes, rivers etc. that can significantly affect the outcome – this type of an uncertainty source is not considered in our simulations.





## 4.2 Simulation uncertainty sources

The overall uncertainty associated with the management actions differs for Hyytiälä and Sodankylä (Fig. 1). This is due to the more abundant harvesting effect at Hyytiälä (Fig. 3), whereas most of the biomass in Sodankylä is left to grow. Sodankylä stand volume increases as simulations progress whereas Hyytiälä stand volume remains the same or even decreases for the BAU scenario. This underlines the importance of proper forest management, as the impact of these relatively similar actions is strong – especially when taken into context of e.g. clear cuts.

As expected, the uncertainty related to the RCP scenarios increases systematically (Ruosteenoja et al., 2016) for all ecosystem indicators and grouped variables (except for the Water group) as the simulations advance further in time. This is similar to results by Kalliokoski et al. (2018). The RCP scenarios are the most dominant factor in explaining the JSBACH and PREBAS uncertainties for both sites at the end of the simulations. The RCP uncertainty also dominates the Carbon, Growth and Snow variables at both sites and Biomass variables for Sodankylä. The RCP scenarios tend to gain effect at mid-century (e.g. Fig. 3), although there are some earlier affects, e.g. snow variables for Sodankylä (Fig. 6).

The effect of the climate models to the redundancy indices is the most varied among the examined uncertainty sources. The climate models tend to have more impact in the two earlier periods, although the overall climate model uncertainty remains roughly the same throughout the simulations. This can be seen as arising from the internal variability of the climate system (Knutti and Sedláček, 2012). The combined variation of climate models and model parameters may not be fully captured due to non-linearity within the simulated variables. This is noted to emphasise the importance of the parameter uncertainty, as stated by Reyer et al. (2016). The parameter uncertainty is expected to be small when compared to the selected RCP scenarios that have a significant impact on the ecosystem (see Holmberg et al., 2019, Fig. 2). Most of the examined parameter distributions are highly alike for all climate models (Fig. 4), especially during the reference period (Fig. 7). The combined climate model and parameter uncertainty is on par with the RCP scenario uncertainty towards the end of the simulations (Fig. 1).

## 4.3 Validity of estimates

The JSBACH model calibration (Mäkelä et al., 2019) was originally used in comparison of various submodel components (stomatal conductance functions) and the PREBAS calibration (Minunno et al., 2019) utilised permanent growth and yield experiments. Both of these examinations rely on hindcasting with relatively recent meteorological measurements or datasets, and the resulting parameter distributions emulate the current climate conditions well. The examined ecosystem indicators were also chosen to reflect the calibrated parameters and processes.

The CCA analysis and model comparison focuses on the relative differences in the ecosystem indicators, and thus less importance is given to the absolute indicator values. The CCA analysis only accounts for linear dependencies (Hotelling and Pabst, 1936) between the input and output uncertainties, and even though the redundancy index (Stewart and Love, 1968) considers the (correlated) variance between the variables, the nonlinear effects may be underestimated. We reduce the annual variation and linearise the variables by averaging and separating them into four consecutive 30-year long periods. Additionally,





we also examined the PREBAS redundancy indices without the RCP2.6 – these results differ only marginally from those with
310 the RCP2.6 included, which increases the validity of the JSBACH results.

This linearisation may not be enough to capture all variation, as is the case with the JSBACH Water group uncertainties (Fig. 2) and the wide spread of soil moisture values (Fig. 7) and cumulative drought days (Fig. 8). The different parameterisations and climate models have a prominent variation, but due to adverse effects the correlations remain small. For example, the RCP8.5 radically increases precipitation (see Ruosteenoja et al., 2016, Fig. 2) and therefore increases the soil moisture (Fig.
7) and reduces the amount of drought days (Fig. 8). The strength of this effect varies among the climate models, but the model parameterisations still enable even radical increases to the number of drought days. This major source of uncertainty, investigated by e.g. Trugman et al. (2018), is not captured by CCA. However, when the indicators are reasonably correlated (as is the case for most of the presented indicators), the estimates are reliable.

The CCA analysis was performed for indicator groups to ensure robustness of the approach – this was not successful in
every case, as a minimal but systematic difference in Sodankylä reference period harvested volume led to a large management scenario impact (Fig. 2). The situation arises as all of the other indicator values were nearly identical and thus a small systematic change that was relatively large, had high correlation and impact in CCA. This event was not replicated with the other groups.

## 5 Conclusions

Our simulations indicate that the management actions have the greatest influence to simulated ecosystem indicators of climate
change. A similar impact is achieved by the RCP scenarios towards the end of century. The combined climate model and parameter uncertainty is also an important factor for the whole duration of the simulations due to internal variability of the climate system, but these effects can be easily underestimated due to non-linear or adverse effects. The examined uncertainties are comparable for both models.

Long-term measurements and simulations indicate considerable increases to GPP and respiration, with a slightly larger
emphasis respectively for the southern and northern boreal forests. While the effect of management to these variables is linear, the impact on NEE is more complex and would be of interest in further studies. The snow melt is occurring several weeks earlier in all simulations and the length of the snow melting period appears to be decreasing, although the results for Sodankylä are not conclusive. Similarly, the length of the vegetation active period is expected to increase linearly for both sites by a few weeks. Sodankylä soil moisture is expected to increase, while the effects for Hyytiälä are varied. The scenarios do not constrain
the recurrence of drought as the parameterisations enable varied outcomes.

*Code and data availability.* The JSBACH model can be obtained from the Max Planck Institute for Meteorology (MPI-M), where it is available for scientific community under the MPI-M Software License Agreement (http://www.mpimet.mpg.de/en/science/models/license/). The R package (Rprebas), containing the PREBAS model, is available on GitHub (https://github.com/checcomi/Rprebas). The model parameter values and the data used for the CCA analysis and redundancy index calculations are available as supplements to this paper.





## Appendix A: Model parameters

The JSBACH and PREBAS model uncertainties are represented by a set of parameter vectors (available as supplements). The different parameters and their distribution means and deviations are given in Tables A1 and A2. The PREBAS parameter values were evenly sampled from the MCMC chains in Minunno et al. (2019). The JSBACH parameter values were taken from the adaptive population importance sampler (APIS) simulations using the Bethy stomatal conductance formulation in Mäkelä et al. (2019). The bulk of these (100) vectors consists of APIS location parameters at 20 iterations (40 samples), which are complemented by later draws to reflect the sampling process.

**Table A1.** PREBAS model parameter descriptions as in Minunno et al. (Table 1; 2019) with distribution mean and standard deviation.

| Parameter description (units) | | pine $\mu$ | $\sigma$ | spruce $\mu$ | $\sigma$ | birch $\mu$ | $\sigma$ |
|---|---|---|---|---|---|---|---|
| Maintenance respiration rate of foliage (kg(C) kg$^{-1}$(C) yr$^{-1}$). | $m_{\mathrm{F,ref}}$ | 0.2 | 0.003 | 0.2 | 0.005 | 0.3 | 0.061 |
| Maintenance respiration rate of fine roots (as above). | $m_{\mathrm{R,ref}}$ | 0.23 | 0.023 | 0.24 | 0.036 | 0.33 | 0.064 |
| Maintenance respiration rate of sapwood (as above). | $m_{\mathrm{S,ref}}$ | 0.03 | $1.4\times10^{-4}$ | 0.03 | $3.0\times10^{-4}$ | 0.03 | $1.4\times10^{-3}$ |
| Growth respiration rate (as above). | c | 0.29 | 0.005 | 0.25 | 0.023 | 0.24 | 0.027 |
| Leaf longevity (yr). | $\nu_{F,\mathrm{ref}}$ | 4.0 | 0.02 | 9.7 | 0.27 | 1.1 | 0.09 |
| Fine root longevity (yr). | $\nu_{\mathrm{R}}$ | 0.9 | 0.03 | 1.7 | 0.07 | 1.2 | 0.19 |
| Homogeneous extinction coefficient. | $k_{\mathrm{H}}$ | 0.25 | $5.4\times10^{-4}$ | 0.25 | $8.8\times10^{-4}$ | 0.31 | $9.7\times10^{-3}$ |
| Specific leaf area (m$^2$ kg$^{-1}$(C)). | $s_{\mathrm{LA}}$ | 20.0 | 0.036 | 20.1 | 0.072 | 41.0 | 2.94 |
| Parameter relating to reduction of photosynthesis with crown length. | $s_1$ | 0.011 | $6.1\times10^{-4}$ | 0.006 | $9.7\times10^{-4}$ | 0.031 | 0.011 |
| Wood density (kg (C) m$^{-3}$). | $\rho_{\mathrm{W}}$ | 197 | 2.82 | 183 | 2.48 | 226 | 20.9 |
| Ratio of fine roots to foliage. | $\alpha_{\mathrm{Rs}}$ | 180 | 0.18 | 201 | 0.55 | 105 | 4.44 |
| Foliage allometry parameter. | z | 1.8 | 0.020 | 1.7 | 0.001 | 1.9 | 0.012 |
| Ratio of total sapwood to above-ground sapwood biomass. | $\beta_0$ | 1.28 | 0.014 | 1.27 | 0.018 | 1.48 | 0.056 |
| Ratio of mean branch pipe length to crown length. | $\beta_{\mathrm{B}}$ | 0.4 | $4.5\times10^{-4}$ | 0.5 | $8.7\times10^{-4}$ | 0.4 | 0.048 |
| Ratio of mean pipe length in stem above crown base to crown length. | $\beta_{\mathrm{S}}$ | 0.39 | 0.006 | 0.46 | 0.007 | 0.46 | 0.024 |
| Light level at crown base that prompts full crown rise. | $C_{\mathrm{R}}$ | 0.22 | 0.008 | 0.16 | 0.004 | 0.17 | 0.013 |
| Reineke parameter. | $N_0$ | 856 | 3.0 | 1040 | 7.4 | 998 | 68.6 |





**Table A2.** JSBACH model parameter descriptions as in Mäkelä et al. (Table 2; 2019) with distribution mean and standard deviation.

| Parameter description (units) | | $\mu$ | $\sigma$ |
|---|---|---|---|
| Farquhar model maximum carboxylation rate at 25 °C (µmol $(CO_2)$ m$^{-2}$ s$^{-1}$) | $V_{C,max}$ | 42.8 | 1.94 |
| Farquhar model efficiency for photon capture at 25 °C. | $\alpha$ | 0.30 | 0.013 |
| Multiplier in momentum and heat stability functions. | $c_b$ | 4.9 | 0.7 |
| Ratio of unstressed C3-plant internal/external $CO_2$ concentration. | $f_{C3}$ | 0.81 | 0.025 |
| Exponential scaling of water stress in reducing photosynthesis. | $q$ | 0.65 | 0.19 |
| Volumetric soil water content above which fast drainage occurs. | $\theta_{dr}$ | 0.79 | 0.09 |
| Fraction depicting relative surface humidity based on soil dryness. | $\theta_{hum}$ | 0.23 | 0.02 |
| Volumetric soil moisture content at permanent wilting point. | $\theta_{pwp}$ | 0.19 | 0.03 |
| Volumetric soil moisture content, above which transpiration is unaffected. | $\theta_{tsp}$ | 0.43 | 0.1 |
| Fraction of precipitation intercepted by the canopy. | $p_{int}$ | 0.29 | 0.04 |
| Depth for correction of surface temperature for snow melt (m). | $s_{sm}$ | 0.05 | 0.025 |
| Maximum water content of the skin reservoir of bare soil (m). | $w_{skin}$ | $2.7\times10^{-4}$ | $7.3\times10^{-5}$ |
| LoGro-P: memory loss parameter for chill days (days). | $C_{decay}$ | 15.7 | 5.3 |
| LoGro-P: minimum value of critical heat sum (°C d). | $S_{min}$ | 18.0 | 6.4 |
| LoGro-P: maximal range of critical heat sum (°C d). | $S_{range}$ | 189.0 | 49.9 |
| LoGro-P: cutoff in alternating temperature (°C). | $T_{alt}$ | 6.0 | 1.8 |
| LoGro-P: memory loss parameter for pseudo soil temperature (°C). | $T_{ps}$ | 15.8 | 5.3 |

## Appendix B: Calculation of ecosystem indicators

Most of the ecosystem indicators in this paper are directly produced by the models, but few are derived from other variables.

### B1 Drought days

The drought days are calculated as the amount of days when average soil moisture (of the combined 2nd and 3rd soil moisture levels in a 5-layer JSBACH scheme) is below a certain threshold. Only summertime (June, July, August) values are used and the threshold for Hyytiälä was set as the 5th percentile of all soil moisture values during the reference period. This value is approximately 33 % of the soil field capacity in Hyytiälä, which compares well with the parameters $\theta_{tsp}$ and $\theta_{pwp}$ for the Hyytiälä drought period optimisation in (Mäkelä et al., 2019). Thus, the number of dry days is a reasonable measure for



Hyytiälä. We used the same percentile to set a similar value for Sodankylä although the site characteristics differ (different soil compositions and field capacity etc.).

## B2  Vegetation active period

The dates for the start of season (SOS) and end of season (EOS) for the vegetative active period are calculated from simulated daily GPP. First we extracted the value corresponding to the 90th percentile of the daily GPP, from all of the simulations during the reference period, and then multiplied this value by 0.15. The SOS date is considered to be the first day of the year (DOY), when the daily GPP is consistently above this threshold. The consistency here means that, when we consider the daily GPP values, starting from the 30th DOY, to twice as far as the date of the SOS event, the GPP must be above the threshold for at least half of the days. The date for EOS is calculated similarly, when GPP is below the threshold and starting from 230th DOY.

## B3  Snow melting period

The snow depth in model simulation varies on a year-to-year basis. We also encounter some years without any snow cover for Hyytiälä. Hence we first aggregate the snow depth over the model parameterisations and climate model simulations to produce average site and RCP scenario specific time series. This approach yields robust estimates of the snow cover, where the actual time series is smooth enough to allow calculation of the beginning of snow melting period and the first snow free date. We take a similar approach as in Manninen et al. (2019) and fit a sigmoidal function to identify the starting date of snow melt. The snow is considered to have melted, when the snow cover has consistently vanished. This means that there is no snow cover for at least half of the days during $\pm 10$ days of the snow clear date.

## Appendix C:  Canonical correlation analysis

We carried out canonical correlation analysis (CCA) to quantify the impact of the different factors on the ecosystem indicators. These factors are parametric uncertainty (pars), management scenarios (man), climate models (clim) and rcp scenarios (rcp). CCA is a multivariate extension of correlation analysis that allows identifying linear relationships between two sets of variables (Hotelling and Pabst, 1936). It's use is similar to multiple regression but it is more appropriate when there are multiple intercorrelated variables such as model outputs. A more detailed description of CCA is provided in (Stewart and Love, 1968).

We consider two sets of variables, $X$ (the different factors) and $Y$ (ecosystem indicators). These are of dimensions $N_p$ and $N_q$, where $N$ is the number of realisations for the variable and $p, q$ are the number of variables. In CCA we construct the linear composites (called canonical variates) $U_1 = a^T X$ and $V_1 = b^T Y$ by maximising the correlation between them. The composites $U_1$ and $V_1$ form the first pair of canonical variates. The second pair is formed similarly but it is required to be uncorrelated with the first pair (and so forth for the following pairs). The first pair accounts for the highest amount of variance between the two sets of variables and has the highest canonical correlation ($Rc$) – the variance and correlations diminish for each consecutive pair.





The correlations between the individual variables (factors or indicators) and the respective canonical variates are called canonical loadings ($CL$), whereas the correlations with the opposite canonical variate are called canonical cross-loadings ($CcL$). These loadings are needed to summarise the CCA results via the use of the redundancy index ($Rd$) that expresses the amount of variance of a set of variables explained by another set of variables (Stewart and Love, 1968; Weiss, 1972; van den Wollenberg, 1977).

$$Rd_{iv} = \frac{1}{n_i} \left( \sum (CL_{iv}^2) \right) Rc_v^2 \qquad (C1)$$

Above $i$ is a placeholder for one of the two sets of variables, factors ($f$) and ecosystem indicators ($e$); $v$ indicates a canonical variate; $n_i$ is the number of variables in the $i$-th set and $Rc$ are the canonical correlations.

    The square of the canonical loadings expresses the proportion of variance accounted for each variable – computing the average for each variate provides an indication of the overall variability explained by the variate. The squared $Rc$ represents

the variance shared by the canonical variates of the two sets of variables – it is the bridge between the two sets. The redundancy index can be summed up across the canonical variates to have an overall measure of the bi-multivariate covariation of the two sets of variables.

    In our analysis, we wanted to quantify the importance that each factor have on the ecosystem indicator uncertainty ($RdF$). We quantified the redundancy index of the indicators for each canonical variate and then multiplied it by the squared canonical

cross-loadings between factors and variates.

$$RdF_{fv} = Rd_{ev} CcL_{fv}^2 \qquad (C2)$$

$CcL$ represents the proportion of variance shared between the factors ($f$) and the canonical variates of the ecosystem indicators ($e$). The $RdF$ of the different factors can be summed up across the variates to obtain the overall weight that each factor has on the ecosystem indicator uncertainty.

*Author contributions.* J. Mäkelä and F. Minunno are respectively responsible for the JSBACH and PREBAS simulations. J. Mäkelä prepared the manuscript, with contributions from all co-authors, whereas F. Minunno provided the CCA and redundancy index calculations and analysis.

*Competing interests.* The authors declare that they have no conflicts of interest.

*Acknowledgements.* This work has been supported by Jenny and Antti Wihuri Foundation and the Ministry of foreign affairs of Finland
(IBA-ForestFires, PC0TQ4BT-53) as well as the Academy of Finland Center of Excellence (272041), OPTICA (295874), the Strategic Research Council at the Academy of Finland (STN-SOMPA, 312932), ICOS Finland (281255) and MONIMET (LIFE12 ENV/FI/000409).



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
