# Peer review of "Sensitivity of 21st century simulated ecosystem indicators to model parameters, prescribed climate drivers, RCP scenarios and forest management actions on two Finnish boreal forest sites"

_Biogeosciences, 2019_

## Referee Comment (RC1) · Martin Thurner (Referee) · 5 Dec 2019

Mäkelä et al. disentangle different sources of uncertainty regarding the projected changes in a set of carbon and water cycle indicators during the 21st century. They apply the land ecosystem model JSBACH and the forest growth model PREBAS, which allows them to study not only the effects of prescribed climates and representative concentration pathways (RCPs), but also of parameter uncertainties and forest management practices.

This comprehensive analysis sheds light into the impact of uncertainties in projected climate, RCPs, model parameters and, which I find most interesting and novel, harvesting practices on a selection of important ecosystem indicators at two boreal forest

sites in Finland. Although the authors have already put huge efforts into this work, I can see a few issues regarding the applied methods and presentation of results that should be addressed before this work can be published in Biogeosciences. I recommend the authors to address especially the major comments below.

Martin Thurner

Major comments

What are the conclusion and implications of this work? This should be highlighted also at the end of the abstract.

Please describe the parameter selection and sampling in more detail in the methods and/or Appendix A, even if this has been presented in earlier studies. This would make it easier for the reader to understand your work. I am not sure what it means to use 100 parameter vectors (Line 105)? Only 100 combinations in total (this seems insufficient to cover the uncertainty in ca. 20 parameters)? Or do you mean 100 values sampled for each parameter? It is also not clear to me why largely different sets of parameters have been selected for the 2 models (Appendix A)?

Since canonical correlation analysis allows identifying linear relationships between two sets of variables, I wonder if this is the most appropriate approach here, given the non-linearities in the processes determining the investigated indicators? I see that this point is mentioned in the discussion section, but wonder if there is no other, better suited approach that could be applied?

It is unclear to me why some of the indicators are calculated for both models and others either for JSBACH or PREBAS (Table 1)? I understand that some indicators may not be simulated by both models, but for instance biomass and soil carbon should also be available from JSBACH simulations.

The potential impact of uncertainty arising from different process implementations (or missing processes) in the models could be at least discussed.

Minor comments

I am not sure if the term "ecosystem indicators of climate change" is very descriptive for the selected indicators. These are a collection of carbon fluxes / stocks and vegetation and water cycle properties and maybe you could find a more suitable name.

Title: You should mention that this study focuses on boreal forest sites (in Finland)

Line 41: The impact of what?

Line 42: Do you mean future forest productivity in Finland or in general?

Lines 61-73: It would be interesting to get to know the distribution in tree diameter at both sites. This would make it easier to understand the differences in the effect of the forest management scenarios at both sites.

Line 83: Maybe describe the harmonized FMI meteorological data in a bit more detail.

Line 100: Which version of JSBACH do you use? Not sure if Kaminski et al. 2013 is the most appropriate reference for the JSBACH land surface model.

Line 102: Please state here that the parameter distributions are shown in the Appendix.

Line 102: How would the results be affected by coupled model runs?

Line 104: You mean only 1 PFT is present in the study regions?

Line 108: Wouldn't the "model uncertainty" also comprise uncertainty due to model structure (i.e. implementation of processes, missing processes)

Line 123: Do you mean mortality due to competition for resources?

Line 136: These are not only biophysical, but also biogeochemical indicators.

Line 138: Please list in detail how many scenarios have been investigated for each uncertainty component. Maybe a table would help to present this.

Table 1: Why is "gross growth" grouped into the "Biomass" group and how does it differ

[Figure]

from GPP?

Lines 158-164: Please state the values of the redundancy index for the different scenarios (at least in the bar plot in Figure 1) and explain what these values actually mean (what is a typical value range of Rd?)! What does it mean if Rd values for all factors are low (cf. Fig. 2)?

Figure 1: I suppose the colour scheme for the JSBACH simulations is not correct (management scenarios are displayed, although they have not been implemented in JSBACH).

Figure 2: Please use the same colour scheme as in Fig. 1.

Line 179: Please always state if you refer to ecosystem, autotrophic or heterotrophic respiration.

Lines 187-188: Isn't this statement in conflict with your previous finding that the impact of parameter uncertainty on overall uncertainty would be small?

Line 197: Which processes are (not) considered in the models that could lead to an impact of management on the seasonality indicators?

Line 222-223: Why are the model parameterisations responsible for the differences in soil moisture distributions and not rather the climate models?

Figures 7 and 8: A legend could explain which colour refers to which climate model.

Figure 8: Why are cumulative drought days displayed and not the trend in drought days over time? This might make it easier to spot changes in drought days.

Line 254: I suggest using a more informative heading than "Impact to ecosystems".

Line 341: I cannot find the supplementary material?!

---

## Referee Comment (RC2) · Benjamin Felzer (Referee) · 22 Jan 2020

The article by Makela et al. applies two ecosystem models to two boreal pine sites in Finland using 3 RCP scenarios from 5 CMIP5 climate models to determine uncertainties in carbon and water variables due to parameters, climate models, RCP scenarios, and management options (which are only covered in one of the ecosystem models). This is an interesting uncertainty analysis, though the breadth of the work is limited as it only covers two sites within a single biome, rather than more broadly defining uncertainty based on a wider-range of sites or biomes or through extrapolation. While the authors have made clear why their choice of particular climate models, it is not clear how these two ecosystem models fit into the broader range of such models. The significance of this study is also not great without further justification. Are these sites
representative of other boreal forested site? Do results for pines also apply to spruce or larch forests? Are the same climatic trends also occurring at most other boreal forested sites? Does one set of model results for two sites really make the case that management is more important than the other factors? The paper can be improved if this approach and results can be put into the broader context.

1. Line 104 (Section 2.3): Please provide more description of the "100 vectors". I count around 20 parameters from Table 1 in the 2019 paper – are these the parameters that are being varied? There needs to be a more thorough discussion of how the parameter uncertainty is determined. 2. Section 3.1 – define DEL (delayed management scenario) 3. Line 162: Should be Figure 3, not 2 4. Figure 4: I assume the respiration is ecosystem respiration, and not heterotrophic or autotrophic, so please specify. 5. Figure 5: How are the Cfluxes determined – are these the mean or net difference (NEE) of the GPP and ecosystem respiration? Similarly, in Figure 9, how are the Wfluxes determined? 6. Figure 8: Need y axis labels and title and x axis title 7. Lines 201-203 (Section 3.4): What is meant by different model parameterizations here – this should not be confused with parameters as used throughout the paper, as isn't the meaning here different functional relationships in the different models? 8. Figure 10: There is a reference to drought frequency from this figure, so a better (or additional) figure would be one that showed the frequency distribution of droughts between the different models. 9. Line 204: What is A1? 10. Figure 13: This figure is really just comparing the two models (JSBACH and PREBAS). There are references in the discussion to this figure about pathways (lines 259-260) and high variation (lines 260-261), but I don't see that from this figure. Also how is linearly lengthening of VAP shown in Figure 6? 11. Lines 251-252: How do you know how much $CO_2$ has contributed to GPP? That would have to come from modeling or a FACE experiment. 12. Lines 304-306 don't make much sense to me, as the amount of drought days is increasing in Figure 10. 13. Conclusions: I like general points that management is most important to carbon, that effects of RCP are more important with time, and that NEE is complex due to offsetting effects of GPP and respiration. But the last paragraph is a weak ending

– can use a better closing paragraph to highlight the importance of the study, how it informs the broader modeling community, and where to go from here.

---

## Referee Comment (RC3) · Florian Hartig (Referee) · 29 Jan 2020

Mäkelä et al. quantify the contribution of different uncertainty sources (climate, RCP, management, model parameters) on uncertainty in predictions of various ecosystem indicators made by two ecosystem models throughout the 21st century.

The topic of the study is timely, and relevant for the field. It also fits well to Biogeosciences. I believe that the study is overall sound. More effort, however, could be spent on discussing the implications of the results for vegetation modeling and climate change research. I also had some questions regarding the quantification of input uncertainties and some other aspects of the methodology (see details below), which should be considered by the authors.

=== General comments ===

1) The abstract does a fine job at describing the methods and results, but the motivation and the implications could be better worked out. The same point could also be made about the main part of the paper - relevance / insight could be better worked out in introduction / conclusion.

2) The way input uncertainties were quantified should be more systematically described and justified. Taking the example of management: you consider two management scenarios, which you effectively consider equally likely. On which basis were those chosen? I assume that there are more options for management. Regarding the RCPs – your analysis seems to put equal weights on all 3 RCPs, so you consider them equally likely? It's OK if your uncertainty ranges are chosen "ad hoc", but it should be clear if you interpret those as a probabilistic, or just as a range of options. It seemed to me that in several parts of the discussion, you interpreted the results more like a sensitivity analysis (as in: through management, we can change a lot) than an uncertainty analysis.

3) Maybe I missed it, but you never explained the reason for using 2 models – I'm also asking because the model (structure) could of course also be seen as a source of uncertainty, but it seems you don't view it that way?

4) I appreciate that there is no ideal method to attribute back the output uncertainty to the inputs, but the CCA certainly has some limitations due to its linearity assumptions. In https://www.sciencedirect.com/science/article/abs/pii/S0378112717304371, we use a random forest for the same problem. There are some caveats for this approach as well, in particular if there is collinearity between input variables, but it might be a more robust alternative if nonlinearity is an issue.

Specific comments

THE forest -> consider deleting "the"
ongoing -> redundant?

The logical connection between the increase of CO2 and the necessity to do this study was not clear to me. Aren't the changes you are analyzing here mostly driven by climate change (which is of course created by CO2 increase)?

Stages – you mean points in time?

indicators of climate change -> I wouldn't say that these are climate change indicators, maybe just "ecosystem indicators"?

This sounds weird – do you mean: the uncertainty induced by the climate model . . .

One would usually expect some kind of conclusion / summary at this point delete "and"

I think this paragraph would be easier to understand if you would start with the topic, which is that there is agreement that climate is changing, but uncertainty about the magnitude

Again, motivation not quite clear to me.

How can they have good performance and at the same time represent well the variation of models? This seems to be a contradiction

Overall, the explanation about the selection of the climate models didn't sound particularly convincing to me. Why would it be scientifically beneficial to have climate models from different continents? The only concern is that the selection you make is not representative of the distribution of climate models as a whole. If you would shortly state that this is not the case, that would fine for me

I would recommend active voice: we used the model . . . for these parts

You mean the "parameter uncertainty"? Because model uncertainty includes more

The parameter description is hard to understand. Also, I assume what you did is to have these 100 parameter combinations, and then you cross them with the other options in a full factorial design? This could be better explained, and also why you choose to do this, as opposed to drawing parameter values from the posterior for al model runs.

Parameter uncertainties?

I think it should have been called like this all the time, because model uncertainty could be read as structural uncertainty

One could think about introducing a subsection explaining and comparing the 4 sources of uncertainty here

Why 2000 vs. 6000? Maybe explain how the numbers come about.

Well, the uncertainty has to be caused by something, right?

I think it would be good to shortly explain the interpretation of the RD index here or before. The reader should be able to interpret the figures without referring to the appendix

Maybe some introductory sentence here that explains why we next look at this?

Fig. 3 Although I understand the reason why you present it like this, it is slightly confusing to have the site on the y axis, and the variable as title of the figure

Not sure if bifurcation is really the best word for it. Just because it is used so often in a slightly different context in maths

I was struggling to see the motivation for this section, in the context of uncertainty – isn't this all more about describing climatic trends?

well, given that the model is (I think deterministic), it must be explained by the input uncertainties. It seems to be rather that there is no linear correlation that could be picked up by the CCA?

[Figure]

Fig. 9 the site labels are very hard to spot

Similar, really? The two models look quite different to me in Fig. 1. Also, Fig. 9 looks to me as if there are quite some differences between the models

It seems to me that you interpret the results here more like a sensitivity analysis than an uncertainty analysis. In an UA, we wouldn't have the choice to change management, it's uncertain.

I'm a bit surprised – are these models suitable to understand changes in snow melting periods? Of so, by what mechanism would that occur, increased LAI?

Again, this would seem to me an interpretation of a SA, not UA

next 4 lines: I didn't understand what you mean here

I don't understand how you can deduce this from Fig. 4 / 7 – isn't the KDE showing the combined other uncertainties, not just the parametric?

when they are correlated, they could also be both wrong in the same way

---

## Author Comment (AC1) · 13 Mar 2020

We have created a pdf file containing all referee comments and our responses to them. Some of the answers were written quite a while ago, so there may be some overlap. Hopefully the answers are satisfactory as they are. Unfortunately I am not able to attach the answers here, because the portal does not seem to allow a large pdf supplement. I've sent the pdf to the copernicus office (and the editor) and hopefully they will upload the file here (I'm trying to avoid copy-pasting the answers here, because that would remove all styles etc. and mean more work).

kind regards and many thanks to the referees for their comments

-JM

---

## Author Comment (AC3) · 18 Mar 2020

The comment was uploaded in the form of a supplement:
https://www.biogeosciences-discuss.net/bg-2019-395/bg-2019-395-AC3-supplement.pdf
* * *

---

## Author Response (AR1)

This document contains the referee comments (indented and with blue font) and our responses, regarding the submission of bg-2019-395: "Uncertainty sources in simulated ecosystem indicators of the 21st century climate change". We have also included the marked up differences (latexdiff) made to the manuscript.

**Referee 1 comments**

Mäkelä et al. disentangle different sources of uncertainty regarding the projected changes in a set of carbon and water cycle indicators during the 21st century. They apply the land ecosystem model JSBACH and the forest growth model PREBAS, which allows them to study not only the effects of prescribed climates and representative concentration pathways (RCPs), but also of parameter uncertainties and forest management practices.

This comprehensive analysis sheds light into the impact of uncertainties in projected climate, RCPs, model parameters and, which I find most interesting and novel, harvesting practices on a selection of important ecosystem indicators at two boreal forest sites in Finland. Although the authors have already put huge efforts into this work, I can see a few issues regarding the applied methods and presentation of results that should be addressed before this work can be published in Biogeosciences. I recommend the authors to address especially the major comments below.

**Major comments**

A. What are the conclusion and implications of this work? This should be highlighted also at the end of the abstract.

We have now added a concluding sentence to the end of the abstract to highlight the importance of management actions. We also note in the abstract that parameter uncertainty is usually not included in these types of simulations. We have also slightly modified the Conclusions section to better reflect this comment and added a new concluding paragraph.

B. Please describe the parameter selection and sampling in more detail in the methods and/or Appendix A, even if this has been presented in earlier studies. This would make it easier for the reader to understand your work. I am not sure what it means to use 100 parameter vectors (Line 105)? Only 100 combinations in total (this seems insufficient to cover the uncertainty in ca. 20 parameters)? Or do you mean 100 values sampled for each parameter? It is also not clear to me why largely different sets of parameters have been selected for the 2 models (Appendix A)?

The parameter selection was based on previous calibration experiments, referenced in the manuscript. These were run prior to the uncertainty simulations (and independently of those). The calibrations were later deemed reasonable for the uncertainty simulations (no new parameters were added as their interdependencies would not be known). The calibration simulations were done independently, so naturally the sets differ (as the focus of the calibrations was different). This explanation also touches on Major comment D and we have made a note of the explanation to the end of the Introduction.

The parameter values were systematically drawn from the calibration processes, so they are not random samples from the predictive distribution. Each sample has been evaluated at some part of the calibration process. For PREBAS, a standard approach was used and values from the MCMC chains were drawn at fixed intervals – this results in an approximation of the posterior distributions, with parameter interdependencies intact. For JSBACH the situation is a bit different as the calibration was done with adaptive population importance sampler (APIS). APIS produces a posterior estimate at each iteration. The estimate at 20-iterations was used as basis and complemented with

later draws from 40, 60, 80 and 100 iterations. This was done to ensure robustness of the posterior estimate.

The parameter vector just means a vector in the parameter space, so it has a value for each parameter. This results in 100 values for each parameter, but they are given in combinations with regards to one another (so the interdependencies are not lost and these are not random samples). We have modified the corresponding text paragraphs (mainly in the Appendix A) to better reflect the referees requests.

C. Since canonical correlation analysis allows identifying linear relationships between two sets of variables, I wonder if this is the most appropriate approach here, given the non-linearities in the processes determining the investigated indicators? I see that this point is mentioned in the discussion section, but wonder if there is no other, better suited approach that could be applied?

The problem here is multiple-in-multiple-out variable dependencies and there are two "commonly" used ways to determine the relationships between two multivariate sets of random variables. These are CCA and analysis of variance (ANOVA) methods, but both of these should yield similar results. CCA has the advantage of exploring multiple outputs together and to better summarise the results (as we have done). CCA should be viewed as a method to (somewhat indefinitely) assess the factors and not a exact measure of the sensitivity of the output.

D. It is unclear to me why some of the indicators are calculated for both models and others either for JSBACH or PREBAS (Table 1)? I understand that some indicators may not be simulated by both models, but for instance biomass and soil carbon should also be available from JSBACH simulations.

The choice of which indicators to calculate was made based on the pre-existing calibration simulations. The parameter calibration etc. was already previously done and the indicators were chosen to represent the most prevalent processes etc. related to the calibrations. With regarding JSBACH, the non-essential variables were removed from the model I/O to reduce runtime. Thus we did not output and save all normally produced variables. Additionally, we did not utilise any "disturbances" for JSBACH, so GPP is a sufficient proxy for biomass. We have added a note of this to the end of the Introduction.

E. The potential impact of uncertainty arising from different process implementations (or missing processes) in the models could be at least discussed.

We have now added a short note of the missing processes in "Validity of estimates" section: "Model calibration and the parameter distributions also compensate for missing and imperfectly modelled processes" as well as touching on the subject in materials and methods section and conclusions. The missing processes are partially accounted for by the different model parameterisations and we believe that the suggested topic would not benefit the manuscript in a meaningful way. Another approach would be to examine model ensembles, which we have only two.

**Minor comments**

1. I am not sure if the term "ecosystem indicators of climate change" is very descriptive for the selected indicators. These are a collection of carbon fluxes / stocks and vegetation and water cycle properties and maybe you could find a more suitable name.

The term "ecosystem indicators" was coined as an overall "umbrella" term to encompass all of the presented indicators. It is clear that the term itself covers also many other indicators, not just the

ones discussed in the manuscript. Unfortunately, we have not been able to come up with an alternative that would be concise and precise.

**2. Title: You should mention that this study focuses on boreal forest sites (in Finland)**

This information has been added to the title.

**3. Line 41: The impact of what?**

The impact of management practices. This has been added to the manuscript.

**4. Line 42: Do you mean future forest productivity in Finland or in general?**

In Finland, this has now been specified in the manuscript.

5. Lines 61-73: It would be interesting to get to know the distribution in tree diameter at both sites. This would make it easier to understand the differences in the effect of the forest management scenarios at both sites.

Unfortunately this information has been lost. When we submitted the manuscript, we informed the editor that PREBAS simulation outputs were lost (only the examined periodic indicator values were recovered) - this is also the reason why we only include more detailed JSBACH images.

**6. Line 83: Maybe describe the harmonized FMI meteorological data in a bit more detail.**

The original word order has been a bit misleading. What was meant is that the reference is FMI meteorological data set, harmonised by Kriging (with external drift). This has now been stated explicitly in the manuscript.

**7. Line 100: Which version of JSBACH do you use? Not sure if Kaminski et al. 2013 is the most appropriate reference for the JSBACH land surface model.**

The JSBACH version (branch: cosmos-landveg-tk-topmodel-peat, revision: 7384) was modified to e.g. include multiple stomatal conductance formulations and delayed effect of temperature to photosynthetic activity in spring. The modifications and the calibrations are described in Mäkelä et. al (2019). The modified model modules (fortran code) is available from github (after agreeing to the MPI license agreement). We agree that the Kaminski reference is not really appropriate, but there are not very good alternatives. We have substituted the reference with two others (Raddatz 2007 and Reick 2013). The model version information has been added to "Code and data availability" section.

**8. Line 102: Please state here that the parameter distributions are shown in the Appendix.**

Added.

**9. Line 102: How would the results be affected by coupled model runs?**

This is a very broad (albeit interesting) question and unfortunately way out of scope of this manuscript, hence we will only give some comments on the topic. Firstly, we are using a model version that was calibrated offline (uncoupled), so the simulations are "in-line" with the calibration. Secondly, coupled model (MPI-ESM) would use a largely different driving data and the setup would be more akin to different MPI-ESM (CMIP) scenarios, not different "climates" *per se*.

**10. Line 104: You mean only 1 PFT is present in the study regions?**

No. In the study regions there are other PFTs, such as birch and understory, but the sites are extensively dominated by evergreen trees. In the simulations (and in the previous calibrations), the

non-evergreen vegetation has been set to occupy a zero fraction of the grid-cell, so there is no other vegetation besides evergreen trees. This has now been stated in the manuscript.

**11.** Line 108: Wouldn't the "model uncertainty" also comprise uncertainty due to model structure (i.e. implementation of processes, missing processes)

This is absolutely true and we have amended the description accordingly.

12. Line 123: Do you mean mortality due to competition for resources?

Mortality is based on the Reineke self-thinning model (Reineke 1933). Reineke, L., 1933. Perfecting a stand-density index for even-aged forests. Retrieved from. J. Agric. Res., Washington.

13. Line 136: These are not only biophysical, but also biogeochemical indicators.

Added.

14. Line 138: Please list in detail how many scenarios have been investigated for each uncertainty component. Maybe a table would help to present this.

We have now added a new Table 1 indicating the number of uncertainty components. The models are run with each combination of these components.

**15.** Table 1: Why is "gross growth" grouped into the "Biomass" group and how does it differ from GPP?

The naming here has been a bit unfortunate, but follows that used in forest sciences. In this instance gross growth refers to increment in tree volume that also considers living, dead and harvested trees. The units in Table 1 for this variable were wrong and have also been corrected.

16. Lines 158-164: Please state the values of the redundancy index for the different scenarios (at least in the bar plot in Figure 1) and explain what these values actually mean (what is a typical value range of Rd?)! What does it mean if Rd values for all factors are low (cf. Fig. 2)?

The value of the redundancy index is between 0 and 1. There are no generally accepted guidelines for the interpretation of these values, hence our interpretation is based on the relative uncertainties between the factors. This is also one of the reasons we did not present the exact Rd values. As an alternative to adding the values in the bar plots, we would suggest presenting them in a Table format as a supplement.

We updated the descriptions in the manuscript accordingly and also added more details in the CCA Appendix so that it may be easier to follow what is actually calculated. If the Rd values for all factors are low, then likely there is low correlation and small variance.

**17.** Figure 1: I suppose the colour scheme for the JSBACH simulations is not correct (management scenarios are displayed, although they have not been implemented in JSBACH).

Yes, this was my error as I forgot to link the updated file to the pdf. This has now been corrected.

**18.** Figure 2: Please use the same colour scheme as in Fig. 1.

Again, as above and has been corrected.

**19.** Line 179: Please always state if you refer to ecosystem, autotrophic or heterotrophic respiration.

In almost every case this is the ecosystem respiration. The CCA analysis for PREBAS was done using autotrophic respiration as indicated in Table 2.

**20.** Lines 187-188: Isn't this statement in conflict with your previous finding that the impact of parameter uncertainty on overall uncertainty would be small?**

This is a question about interpretation. The variation of the parameter values (the actual "spread" of the value distributions) is not in itself parameter "uncertainty". The parametrisations represent different realisations of the model (same as e.g. different RCP's). Parameter uncertainty should be examined as the difference between the "initial spread" and the "end spread". Furthermore, the effect of climate models and RCP scenarios should also be removed. When these are taken into account, the "initial spread" and "end spread" are relatively alike. This is a bit simplified explanation, but should suffice here.

So the uncertainty is not directly the spread of the values, but rather how (and if) the distribution changes over time (since our interest is on the relative changes arising from the different factors). This shorter explanation has been amended and added to the manuscript.

**21.** Line 197: Which processes are (not) considered in the models that could lead to an impact of management on the seasonality indicators?**

Firstly, management affects e.g. the amount of trees and therefore LAI, transpiration, albedo, soil water content, soil carbon content etc. The question about which missing processes could impact management is a bit more difficult. Ingrowth (i.e., the volume of young trees that enter to the measurable size classes) is not implemented in PREBAS. Also understory is not modelled. However, in managed forests, these processes are minimal and the lack of them in the modelling framework should not affect the analysis.

**22.** Line 222-223: Why are the model parameterisations responsible for the differences in soil moisture distributions and not rather the climate models?**

The text has been poorly worded. It was meant as an observation that the soil moisture value distributions are similar for all climate models during the reference period, but the same distributions differ for the last 30-years. The value spread is produced by 100 simulations for each climate model, where only the parameters vary. The observed differences are due to both climate models and parameters (and we can also see differences due to RCP scenarios). We have amended the text accordingly.

**23. Figures 7 and 8: A legend could explain which colour refers to which climate model.**

Yes, the legend was originally left out as we did not want the discussion to devolve into a comparison of individual climate models. This does not seem to be an issue in the manuscripts current shape, so we will add the legends to the figures.

**24. Figure 8: Why are cumulative drought days displayed and not the trend in drought days over time? This might make it easier to spot changes in drought days.**

There is a lot of year-to-year variation in the annual number of drought days (from 0 to 89 days, with 30-year deviation typically ranging from 6 to 16 days), which makes the "trend plot" visually extremely challenging. This also means that the correlations in such plots are negligible (some positive, few negative but mostly no trends according to Mann-Kendall - we have added this information to the manuscript). Below is periodically and over climate-model specific simulations averaged plot of drought recurrence. For these individual models we can fit trendlines, but the Mann-Kendall test does not yield a trend for all simulations (in any of the four cases below).

Climate model averaged annual recurrence of drought [days]

25. Line 254: I suggest using a more informative heading than "Impact to ecosystems". Changed to "Ecosystem indicator sensitivity".

**26. Line 341: I cannot find the supplementary material?!**

We will make the supplements available at the next opportunity.

**Referee 2 comments**

The article by Makela et al. applies two ecosystem models to two boreal pine sites in Finland using 3 RCP scenarios from 5 CMIP5 climate models to determine uncertain-ties in carbon and water variables due to parameters, climate models, RCP scenarios, and management options (which are only covered in one of the ecosystem models). This is an interesting uncertainty analysis, though the breadth of the work is limited as it only covers two sites within a single biome, rather than more broadly defining uncertainty based on a wider-range of sites or biomes or through extrapolation. While the authors have made clear why their choice of particular climate models, it is not clear how these two ecosystem models fit into the broader range of such models. The significance of this study is also not great without further justification. Are these sites representative of other boreal forested site? Do results for pines also apply to spruce or larch forests? Are the same climatic trends also occurring at most other boreal forested sites? Does one set of model results for two sites really make the case that management is more important than the other factors? The paper can be improved if this approach and results can be put into the broader context.

It appears that the referee has got a hold of a previous version of the manuscript (bg-2019-395-manuscript-version1.pdf) and not the preprint version available at the discussion page. We have tried to answer these comments while reflecting both manuscript versions.

Line 104 (Section 2.3): Please provide more description of the "100 vectors". I count around 20 parameters from Table 1 in the 2019 paper – are these the parameters that are being varied? There needs to be a more thorough discussion of how the parameter uncertainty is determined.

This comment is in line with major comment B of referee 1. We have added information regarding the parameter sampling etc. to the manuscript text and appendix A.

2. Section 3.1 – define DEL (delayed management scenario)

The delayed ecosystem logging (DEL) is now defined in Section 2.4 while introducing the PREBAS model, but as a reminder we have also added these definitions in Section 3.1.

3. Line 162: Should be Figure 3, not 2

We gathered all the CCA images to one figure and this reference has been corrected.

4. Figure 4: I assume the respiration is ecosystem respiration, and not heterotrophic or autotrophic, so please specify.

Yes, this has now been stated (at other parts of the manuscript as well).

5. Figure 5: How are the Cfluxes determined – are these the mean or net difference (NEE) of the GPP and ecosystem respiration? Similarly, in Figure 9, how are the Wfluxes determined?

These variables are defined in Table 1 and the uncertainty is extracted from all variables simultaneously. These include GPP, NPP, NEE and respiration as well as soil carbon for PREBAS.

6. Figure 8: Need y axis labels and title and x axis title

This information was previously in the image title, but has now been relocated as axis labels.

7. Lines 201-203 (Section 3.4): What is meant by different model parameterizations here – this should not be confused with parameters as used throughout the paper, as isn't the meaning here different functional relationships in the different models?

This section has been amended in the new version. What was meant here is that the parameterisations, together with the drivers (climate models), yield initially similar value distributions but the situation is different when we examine the last 30 years.

8. Figure 10: There is a reference to drought frequency from this figure, so a better (or additional) figure would be one that showed the frequency distribution of droughts between the different models.

The problem with drought frequency is that it varies considerably throughout the years. Hence, we cannot plot any trendlines etc. for this kind of plot. We circumvented the issue by examining the accumulated drought days. Please also see our answer to question 24 by referee 1.

**9. Line 204: What is A1?**

This was a reference to Appendix A1 and it has now been explicitly specified.

10. Figure 13: This figure is really just comparing the two models (JSBACH and PREBAS). There are references in the discussion to this figure about pathways (lines 259-260) and high variation (lines 260-261), but I don't see that from this figure. Also how is linearly lengthening of VAP shown in Figure 6?

The pathways in this context refer to the development NEE through time with different management actions. The BAU scenario follows a convex "path" and DEL a concave one and these "paths" are separate in Fig. 10. Moreover the time development is "continuous" from left to right along the path. In Fig. 6 VAP is the difference between SOS (yellow) and EOS (red) - the development of both of these is linear and therefore it is linear for VAP as well.

11. Lines 251-252: How do you know how much CO2 has contributed to GPP? That would have to come from modeling or a FACE experiment.

This was a result from Grönholm et. al (2018), but since the reference only directs to an abstract and they have not yet submitted/published the results (that were presented in EGU2018), we have also added a global article reference to reaffirm the credibility of the claim.

**12.** Lines 304-306 don't make much sense to me, as the amount of drought days is increasing in Figure 10.

Figure 10 shows the amount of cumulative drought days from the beginning of 1980. Therefore the recurrence of drought remains the same if the cumulative drought days are linearly increasing. WE have modified the image heading to highlight that the drought days are accumulated from the beginning of 1980. Please also see our answer to question 24 by referee 1 - this explains why we are not using annual drought days (the variation if too large for the image to make sense).

13. Conclusions: I like general points that management is most important to carbon, that effects of RCP are more important with time, and that NEE is complex due to offsetting effects of GPP and respiration. But the last paragraph is a weak ending – can use a better closing paragraph to highlight the importance of the study, how it informs the broader modeling community, and where to go from here.

We have now added a new concluding paragraph.

**Referee 3 comments**

Mäkelä et al. quantify the contribution of different uncertainty sources (climate, RCP, management, model parameters) on uncertainty in predictions of various ecosystem indicators made by two ecosystem models throughout the 21st century. The topic of the study is timely, and relevant for the field. It also fits well to Biogeosciences. I believe that the study is overall sound. More effort, however, could be spent on discussing the implications of the results for vegetation modeling and climate change research. I also had some questions regarding the quantification of input uncertainties and some other aspects of the methodology (see details below), which should be considered by the authors.

**General comments**

A. The abstract does a fine job at describing the methods and results, but the motivation and the implications could be better worked out. The same point could also be made about the main part of the paper - relevance / insight could be better worked out in introduction / conclusion.

Hopefully the changes made in the manuscript are sufficient.

B. The way input uncertainties were quantified should be more systematically described and justified. Taking the example of management: you consider two management scenarios, which you effectively consider equally likely. On which basis were those chosen? I assume that there are more options for management. Regarding the RCPs – your analysis seems to put equal weights on all 3 RCPs, so you consider them equally likely? It's OK if your uncertainty ranges are chosen "ad hoc", but it should be clear if you interpret those as a probabilistic, or just as a range of options. It seemed to me that in several parts of the discussion, you interpreted the results more like a sensitivity analysis (as in: through management, we can change a lot) than an uncertainty analysis.

This comment is much appreciated as much of our interpretation is more akin to sensitivity analysis. We have now updated the manuscript title and give a more overall view of the experiment design (and reasons for the choices made) at the beginning of the "materials and methods" section. Even though much of the discussion is in the style of sensitivity analysis, we have sticked to our original uncertainty terminology and give the justification as per Swart et. al (2009) [Swart, R., Bernstein, L., Ha-Duong, M., and Petersen, A.: Agreeing to disagree: uncertainty management in assessing climate change, impacts and responses by the IPCC, Climatic Change, 92, 1–29, 10.1007/s10584-008-9444-7].

C. Maybe I missed it, but you never explained the reason for using 2 models – I'm also asking because the model (structure) could of course also be seen as a source of uncertainty, but it seems you don't view it that way?

We have now included this information in the overview at the beginning of "materials and methods" section. The model structure is, of course, a source of uncertainty but it is (at least partially) compensated by model calibration and the use of an ensemble of parameter values (instead of single point estimate). This topic is slightly outside the design of our experiment and although we could speculate how the model deficiencies affect the relative uncertainty estimates, there is not much that can be said with any (reliable) amount of certainty.

**D.** I appreciate that there is no ideal method to attribute back the output uncertainty to the inputs, but the CCA certainly has some limitations due to its linearity assumptions.

In https://www.sciencedirect.com/science/article/abs/pii/S0378112717304371, we use a random forest for the same problem. There are some caveats for this approach as well, in particular if there is collinearity between input variables, but it might be a more robust alternative if nonlinearity is an issue.

We appreciate the suggestion but at this point we are not changing the methods. As remarked earlier, some of our interpretation is more in the line of sensitivity analysis and we have not quantified the uncertainties exactly. Even though CCA has not captured the variation in the outputs, we have shown that all inputs have an impact on the ecosystem indicators (namely Water cycle variables) and considering the nature of the manuscript, we believe that this is enough. However, we have added the suggested method as a reference in the conclusions so that others who might consider using CCA have another option as well.

**Specific comments**

1. L1 THE forest -> consider deleting "the"

Considered and deleted.

2. L2 ongoing -> redundant?

Yes. We have now removed the offending word.

**3.** L4 The logical connection between the increase of CO2 and the necessity to do this study was not clear to me. Aren't the changes you are analyzing here mostly driven by climate change (which is of course created by CO2 increase)?

Absolutely correct. We have amended the first sentence in the abstract to better highlight CO2 as a driver of climate change.

4. L8 Stages – you mean points in time?

Yes. We have amended the sentence.

5. L9 indicators of climate change -> I wouldn't say that these are climate change indicators, maybe just "ecosystem indicators"?

Modified all "ecosystem indicators of climate change" to -> "ecosystem indicators".

6. L11 This sounds weird – do you mean: the uncertainty induced by the climate model...

Yes, amended.

7. L14 One would usually expect some kind of conclusion / summary at this point

We have now added here a more impactful statement about the management.

8. L16 delete "and"

Sentence was modified.

**9.** L16 I think this paragraph would be easier to understand if you would start with the topic, which is that there is agreement that climate is changing, but uncertainty about the magnitude

We have included the magnitude uncertainty as the second sentence.

10. L24 Again, motivation not quite clear to me.

Hopefully the previous addition and other changes in the paragraph clarify the motivation.

11. L87 How can they have good performance and at the same time represent well the variation of models? This seems to be a contradiction

The models have good performance in terms of simulating current climate whereas the statement on variation refers to their performance in predicting the future and serves as an introduction on the more elaborated description on the performance that follows. The latter sentence has been deleted and the following sentence slightly reworded in order to bridge from current day considerations to the future.

12. L98 Overall, the explanation about the selection of the climate models didn't sound particularly convincing to me. Why would it be scientifically beneficial to have climate models from different continents? The only concern is that the selection you make is not representative of the distribution of climate models as a whole. If you would shortly state that this is not the case, that would fine for me

We deleted the sentence about the geographic versatility of the origins of the models. The different aspects regarding the representativeness of the selected models in terms of seasonal temperature and precipitation changes have been discussed in the text with quite some detail already. We believe that by removing the admittedly misleading notion about good representation of the variation (see previous comment) we direct reader's attention better to the description of their performances among the 24 CMIP5 models used in the analysis of Ruosteenoja et al (2016).

13. L101 I would recommend active voice: we used the model...for these parts

Changed.

14. L105 You mean the "parameter uncertainty"? Because model uncertainty includes more

Yes, we have now amended the manuscript and systematically refer to this as parameter uncertainty.

**15.** L105 The parameter description is hard to understand. Also, I assume what you did is to have these 100 parameter combinations, and then you cross them with the other options in a full factorial design? This could be better explained, and also why you choose to do this, as opposed to drawing parameter values from the posterior for all model runs.

Yes, we have now included an amended parameter description at the beginning of the "materials and methods" section, where we give an overview of the simulation design.

16. L106 Parameter uncertainties?

This has been amended (see answer above).

17. L109 I think it should have been called like this all the time, because model uncertainty could be read as structural uncertainty

This has been amended, see answer to 14.

**18.** L135 One could think about introducing a subsection explaining and comparing the 4 sources of uncertainty here

We have added this description to the start of "materials and methods" section.

19. L138 Why 2000 vs. 6000? Maybe explain how the numbers come about.

We have added a new Table 1 to the manuscript that explains this.

20. L159 Well, the uncertainty has to be caused by something, right?

Agreed.

21. L160 I think it would be good to shortly explain the interpretation of the RD index here or before. The reader should be able to interpret the figures without referring to the appendix

We have added a description of Rd values etc. in the manuscript, at the end of "materials and methods" section.

22. L166 Maybe some introductory sentence here that explains why we next look at this? Fig. 3 Although I understand the reason why you present it like this, it is slightly confusing to have the site on the y axis, and the variable as title of the figure

We added a sentence at the end of the preceding paragraph. We have also modified (rotated) the images so that Hyytiälä and Sodankylä are presented column wise and the variables on y-axis.

23. L179 Not sure if bifurcation is really the best word for it. Just because it is used so often in a slightly different context in maths

Agreed, the sentence was slightly modified and the "bifurcation" changed to "divergence".

24. L195 I was struggling to see the motivation for this section, in the context of uncertainty—isn't this all more about describing climatic trends?

Partially yes, as explained in answer to general question B, we also consider indicator sensitivity and in such cases the climatic trends are interesting.

**25.** L202 well, given that the model is (I think deterministic), it must be explained by the input uncertainties. It seems to be rather that there is no linear correlation that could be picked up by the CCA?

Absolutely correct, this was meant to be taken in the context of our analysis method. We have amended the sentence.

26. Fig. 9 the site labels are very hard to spot

We have now redrawn the images with bigger labels etc.

27. L249 Similar, really? The two models look quite different to me in Fig. 1. Also, Fig. 9 looks to me as if there are quite some differences between the models

We have clarified the meaning that the uncertainty estimates are mostly similar, especially when we take into account the management effects.

28. L264 It seems to me that you interpret the results here more like a sensitivity analysis than an uncertainty analysis. In an UA, we wouldn't have the choice to change management, it's uncertain.

Yes, this was addressed in general comment B.

**29.** L272 I'm a bit surprised – are these models suitable to understand changes in snow melting periods? If so, by what mechanism would that occur, increased LAI?**

The snow melting was included only for the JSBACH, where the model tracks the accumulated amount of snow and its density, which affects e.g. soil temperature. The mechanisms are quite straightforward and snow melting is due to temperature changes. The changes in snow melting period should therefore be mostly affected by climatic drivers.

**30. L281 Again, this would seem to me an interpretation of a SA, not UA**

Yes, this was addressed in general comment B.

**31. L291 next 4 lines: I didn't understand what you mean here**

We have clarified the meaning.

**32.** L295 I don't understand how you can deduce this from Fig. 4 / 7 – isn't the KDE showing the combined other uncertainties, not just the parametric?

The text was a bit misleading and has been amended (not parameter distributions but value distributions induced by the parameterisations). In Fig 4 the KDE is indeed containing other sources, but it is explained in the text that the distributions are similar for all climate models and RCPs. In Fig. 7 the value distributions are shown for the different RCPs and for each climate model separately.

**33. L317 when they are correlated, they could also be both wrong in the same way**

Yes, the sentence was modified to indicate that the effect is captured by CCA (not that it would be automatically reliable).

**Uncertainty sources in simulated ecosystem indicators Sensitivity of the 21st century simulated ecosystem indicators to model parameters, prescribed climate changedrivers, RCP scenarios and forest management actions on two Finnish boreal forest sites**

Jarmo Mäkelä1, Francesco Minunno2, Tuula Aalto1, Annikki Mäkelä2, Tiina Markkanen1, and Mikko Peltoniemi3

1Finnish Meteorological Institute, P.O. Box 503, FI-00101 Helsinki, Finland

2Department of Forest Sciences, P.O. Box 27, FI-00014 University of Helsinki, Finland 3Natural Resources Institute Finland, P.O. Box 2, FI-00791 Helsinki, Finland

Correspondence: Jarmo Mäkelä (jarmo.makela@fmi.fi), Francesco Minunno (francesco.minunno@helsinki.fi)

Abstract. The forest Forest ecosystems are already responding to increased changing environmental conditions that are driven by increased atmospheric  $CO_2$  concentrations and changing environmental conditions. These ongoing . These developments affect how societies can utilise and benefit from the woodland areas in the future, be it e.g. climate change mitigation as carbon sinks, lumber for wood industry or preserved for nature tourism and recreational activities. We assess the effect and the relative

- 5 magnitude of different uncertainty sources in ecosystem model simulations from the year 1980 to 2100 for two Finnish boreal forest sites. The models used in this study are the land ecosystem model JSBACH and the forest growth model PREBAS. The considered uncertainty sources for both models are model parameters , and four prescribed climates and with two RCP (Representative Concentration Pathway) scenarios. Usually, model parameter uncertainty is not included in these types of uncertainty studies. PREBAS simulations also include an additional RCP scenario and two forest management actionsscenarios. We assess
- 10 the effect of these sources of variation at four different stages of the simulations points in time on several ecosystem indicators<del>of elimate change</del>, 
[revised manuscript text omitted]

| stand volume              | V                  | $\mathrm{m}^3$ / ha             | m 3 / ha |     | Biomass |  |
| harvested volume          | Vharv              | $\mathrm{m}^3$ / ha             |                     | х   | Biomass |  |
| volume of dead trees      | Vmort              | $\mathrm{m}^3$ / ha             |                     | х   | Biomass |  |
| tree biomass              | Biom               | kg(C)                           |                     | х   | Biomass |  |
| tree litterfall           | Lit                | kg(C)                           |                     | х   | Biomass |  |
| leaf area index           | LAI                | $\mathrm{m}^2$ / $\mathrm{m}^2$ |                     | х   | Biomass |  |
| gross growth              | Growth             | $\mathrm{m}^3$ / ha             |                     | х   | Biomass |  |
| gross primary production  | GPP                | $g(C) / m^2 day$                | х                   | х   | Carbon  |  |
| net primary production    | NPP                | $g(C) / m^2 day$                | х                   | х   | Carbon  |  |
| net ecosystem exchange    | NEE                | $g(C) / m^2 day$                | х                   | х   | Carbon  |  |
| respiration (autotrophic) | $R_{at}$           | $g(C) / m^2 day$                |                     | х   | Carbon  |  |
| respiration (ecosystem)   | $\mathbf{R}_{eco}$ | $g(C) / m^2 day$                | х                   |     | Carbon  |  |
| soil carbon               | Csoil              | kg(C)                           |                     | х   | Carbon  |  |
| start of growing season   | SOS                | DOY                             | х                   | Х   | Growth  |  |
| end of growing season     | EOS                | DOY                             | х                   | Х   | Growth  |  |
| length of growing season  | VAP                | days                            | х                   | Х   | Growth  |  |
| evapotranspiration        | ET                 | mm / day                        | х                   | Х   | Water   |  |
| annual soil water         | aSW                | mm                              |                     | Х   | Water   |  |
| summer soil water         | sSW                | mm                              | х                   | Х   | Water   |  |
| number of dry days        | Ddry               | days                            | х                   |     | Water   |  |
| albedo                    | alb                |                                 | х                   |     | Snow    |  |
| snow amount               | snow               | m                               | х                   |     | Snow    |  |
| start of snow melt        | melt               | DOY                             | х                   |     | Snow    |  |
| snow clear date           | clear              | DOY                             | х                   |     | Snow    |  |
| length of snow melt       | SM                 | days                            | х                   |     | Snow    |  |
|                           |                    |                                 |                     |     |         |  |

JSBACH and additionally management scenarios (man) for PREBAS. The indicators were averaged and divided into four consecutive 30-year long periods for both models: 1980-2009 (p1, reference), 2010-2039 (p2, interim), 2040-2069 (p3, midcentury) and 2070-2099 (p4, future). This produced single indicator values for each period and simulation (single instance of each factor) that were calculated for both sites separately. CCA is a multivariate extension of correlation analysis that allows identifying linear relationships between two sets of variables (?). We summarise the CCA results with the use of the redundancy index (Rd) that expresses the amount of variance

185 of in a set of variables explained by another set of variables (ecosystem indicators) by CCA uncertainty factors) (???). In essence, the redundancy index takes into account both correlation and variance between uncertainty factors and ecosystem indicators. The value  $R_d \in [0,1]$ , where a higher value indicates that the factor explains more of the uncertainty related to a given indicator (group). There are no general guidelines for the interpretation of the  $R_d$  values. Therefore, we examine the resulting indices in relation to one another to reveal relative uncertainties. The details of the CCA and the redundancy index

190 are given in appendix Appendix C.

**3 Results**

Forest management was the most dominant factor of uncertainty for Hyytiälä (Fig. 1) throughout the simulation. There was a clear difference for Sodankylä, where management gains only half as much influence. Disregarding management, the climate models and RCP scenarios represent major sources of both JSBACH and PREBAS predictive uncertainty. The impact of climate

195 models was dominant during the reference and interim periods and remained roughly constant over time. The importance of RCP scenarios increased towards the end of the simulations, catching up to management impact at Hyytiälä in mid-century and representing the most important factor during the last period. The parametric uncertainty was the least influential factor for both JSBACH and PREBAS, at both sites. We will next examine the grouped indicator results.

Redundancy indices calculated separately for the different indicator groups.

**200 3.1 Biomass distribution**

205

The site-level differences in biomass stock uncertainties largely arise from the management actions (Fig. 2 and 3) and the management and RCP scenario impacts reflect the redundancy indices calculated with all ecosystem indicators (Fig. 1) for PREBAS. The RCP scenario influence increases for both sites towards the end of the simulations and the climate model and parameter uncertainty is negligible for both sites and all periods. There is an anomaly for Sodankylä reference period, where management has a very large impact. This situation arises due to minimal (0.1 m3/ha), but systematic difference in harvested volume – the difference is so small it is not visually evident (Fig. 3). The rest of the Sodankylä reference period variables are nearly identical, so the small change in harvesting results in high correlation, which is captured by the CCA.

The differences in site-specific variables due to the management actions, can already be seen from the reference period indicators (Fig. 3). The DEL delayed ecosystem logging (DEL) scenario has approximately 10 % larger stand volume than

210 **BAU** business as usual (BAU) for Hyytiälä, but there is practically no difference for Sodankylä. The management actions start to have a noticeable impact for Sodankylä simulated variables at mid-century, but this impact is much smaller than that of the RCP scenarios. The management effect is much more pronounced at Hyytiälä, where both actions follow separate pathways.